 

# The dynamic transmission of positional information in *stau⁻* mutants during *Drosophila* embryogenesis

Zhe Yang[1,2†], Hongcun Zhu[1†], Kakit Kong[1†], Xiaoxuan Wu[1], Jiayi Chen[1], Peiyao Li[1], Jialong Jiang[1], Jinchao Zhao[1], Bofei Cui[1], Feng Liu[1*]

[1]State Key Laboratory of Nuclear Physics and Technology & Center for Quantitative Biology, Peking University, Beijing, China; [2]China National Center for Biotechnology Development, Beijing, China

**Abstract** It has been suggested that Staufen (Stau) is key in controlling the variability of the posterior boundary of the Hb anterior domain ($x_{Hb}$). However, the mechanism that underlies this control is elusive. Here, we quantified the dynamic 3D expression of segmentation genes in *Drosophila* embryos. With improved control of measurement errors, we show that the $x_{Hb}$ of *stau⁻* mutants reproducibly moves posteriorly by 10% of the embryo length (EL) to the wild type (WT) position in the nuclear cycle (nc) 14, and that its variability over short time windows is comparable to that of the WT. Moreover, for *stau⁻* mutants, the upstream Bicoid (Bcd) gradients show equivalent relative intensity noise to that of the WT in nc12–nc14, and the downstream Even-skipped (Eve) and cephalic furrow (CF) show the same positional errors as these factors in WT. Our results indicate that threshold-dependent activation and self-organized filtering are not mutually exclusive and could both be implemented in early *Drosophila* embryogenesis.

**\*For correspondence:**
liufeng-phy@pku.edu.cn

[†]These authors contributed equally to this work

**Competing interests:** The authors declare that no competing interests exist.

## Introduction

During the development of multicellular systems, the expression of patterning genes dynamically evolves and stochastically fluctuates (*Dubuis et al., 2013*; *Gregor et al., 2007a*; *Gregor et al., 2007b*; *Jaeger et al., 2004*; *Kanodia et al., 2009*; *Liu et al., 2013*; *Yang et al., 2018*).The high degree of accuracy (*Dubuis et al., 2013*; *Gregor et al., 2007a*) and robustness (*Houchmandzadeh et al., 2002*; *Inomata et al., 2013*; *Liu et al., 2013*; *Lucchetta et al., 2005*) that developmental patterning achieves is intriguing. Two hypotheses have been proposed to explain these traits: one is the threshold-dependent positional information model, that is the French flag model, which assumes that the positional information is faithfully transferred from precise upstream patterning (*He et al., 2008*; *Wolpert, 2011*; *Gregor et al., 2007a*); the other is the self-organized filtering model, which assumes that noisy upstream patterning needs to be refined to form downstream patterning with sufficient positional information (*Dubuis et al., 2013*; *Houchmandzadeh et al., 2002*; *Jaeger et al., 2004*; *Kanodia et al., 2009*; *Manu et al., 2009*). The two models have often been thought to be mutually exclusive, and which one is implemented in a particular developmental system has been extensively debated. Recently, the two models have also been suggested to collaborate in some developmental patterning systems, but this is still a hypothesis and more molecular-based concrete examples remain to be illustrated (*Green and Sharpe, 2015*).

The *Drosophila* embryo is an excellent model system in which to address this question. The blueprint of the adult *Drosophila* body plan is established during the first 3 hr of patterning before gastrulation in embryos. In particular, the adult body segments can be mapped with the expression pattern of the segmentation genes along the anterior and posterior (AP) axis. The hierarchic

**eLife digest** Broadly speaking, all individuals of any animal species share a highly consistent shape and structure. Despite this, the activity of the genes that control these body patterns can vary significantly. There are currently two models that have been proposed for how noisy systems of genes, and the proteins they code, can produce consistent body patterns. The first, suggests the noise is essentially self-compensating so stably produces the same result, while the second invokes localized self-organizing systems that help to refine the structural details.

In the early stages of development for the fruit fly, *Drosophila melanogaster*, one of the proteins that controls body patterns is called Hunchback (often just Hb for short). The Hb proteins are largely found at the front-end of the fly embryo, with a sharp drop near the middle. Normally the position of the drop in Hb varies between flies by around 1% of the total length of the fly embryo. Previous work has linked a gene called *staufan* (or *stau* for short) to the distribution of Hb in flies but the mechanism involved is unknown.

Yang, Zhu, Kong et al. have now used a technique called light sheet microscopy to accurately measure the location of Hb proteins in fruit fly embryos. Without the *stau* gene, the average position of the drop in Hb proteins underwent a larger shift towards the rear at a key stage in development. Despite this altered behavior, the extent of variation between flies did not change. Similarly, the variation of other genes that control Hb location and that are controlled by Hb remained unchanged. As such, it seems *stau* affects Hb positioning but has no impact on variation between individuals.

These findings suggest that both models for controlling variation in fly development could still be relevant and may operate together. This study also provides a new method for the more precise measurement of systems like these that may offer insights into the mechanisms involved in early embryonic development.

segmentation gene network consists of four layers of patterning genes (*Gregor et al., 2007a*; *Jaeger, 2011*): maternal morphogen such as *bicoid* (*bcd*) (*Liu et al., 2013*; *Porcher and Dostatni, 2010*; *Struhl et al., 1989*), zygotic gap genes such as *hunchback* (*hb*) (*Struhl et al., 1992*), pair rule genes such as *even-skipped* (*eve*) (*Goto et al., 1989*), and segmentation polarity genes (*Swantek and Gergen, 2004*). They form increasingly refined developmental patterns along the AP axis until the positional information carried by the patterning genes reaches the single-cell level, that is 1% of embryo length (EL), in the variability of the expression pattern (*Dubuis et al., 2013*).

The question of which hypothesis is valid in the dynamic transmission of positional information during *Drosophila* embryogenesis has been controversial, and the Hb boundary has been an important subject in these investigations (*Gregor et al., 2007a*; *Houchmandzadeh et al., 2002*; *Huang et al., 2017*; *Lucas et al., 2018*; *Staller et al., 2015*; *Tran et al., 2018*). Lying directly downstream of maternal gradients, zygotic Hb forms a steep posterior boundary in the anterior domain ($x_{Hb}$) on the dorsal side at approximately the middle of the embryo, with a variability of 1% EL. The variability of $x_{Hb}$ has long been thought to depend on the gene *staufan* (*stau*) (*Houchmandzadeh et al., 2002*). As shown in previous work, *stau* was the only gene that dramatically increased the variability of $x_{Hb}$ from 1% EL to more than 6% EL for protein profiles (*Houchmandzadeh et al., 2002*) and 4% EL for mRNA profiles (*Crauk and Dostatni, 2005*). By contrast, the variability of $x_{Hb}$ remains almost the same as that of the wild type (WT) with knockout of nearly all the genes that potentially interact with Hb, including *nos*, *Kr*, and *kni*, or even deletion of the whole or half of the chromosome (*Houchmandzadeh et al., 2002*). Hence, *stau* could be the key to understanding the potential noise-filtering mechanism.

However, the mechanism that underlies of the role of *stau* has been elusive. It is well known that Stau is an RNA-binding protein and it does not interact with Hb directly. But Stau is unique among the maternal factors affecting anteroposterior patterning, as it is necessary for not only the localization of the *bcd* mRNA at the anterior pole (*Ferrandon et al., 1994*; *Struhl et al., 1992*) but also the spatially constrained translation of *nos* mRNA at the posterior pole (*St Johnston et al., 1991*). Hence, Stau is important for establishing the Bcd gradient that activates the transcription of Hb

(*Struhl et al., 1989*) and the Nos gradient that represses the translation of maternal Hb (*Wang and Lehmann, 1991*).

The role of Bcd in regulating the variability of the Hb boundary remains controversial. On the one hand, *Houchmandzadeh et al., 2002* suggested that the variability of $x_{Hb}$ was independent of upstream Bcd gradients, as the average Bcd gradients of two groups of embryos overlapped, although their Hb moved anteriorly or posteriorly compared with the average Hb profile, and the variability of $x_{Hb}$ was much smaller than the positional error derived from the Bcd gradient noise. On the other hand, *He et al., 2008* showed that the shift of $x_{Hb}$ correlated with the average Bcd gradient showing different concentrations at $x_{Hb}$, that the variability of $x_{Hb}$ was equivalent to the positional error derived from the Bcd gradient noise, and that this variability increased when the Bcd gradient profile was altered to show increased flatness toward the mid-embryo (*He et al., 2010*). In addition, the variability of $x_{Hb}$ seemed to be significantly less in He's measurement (*He et al., 2008*) than in Houchmandzadeh's measurement (*Houchmandzadeh et al., 2002*).

Some of these controversial points might be clarified if we could further reduce the spatial and temporal measurement errors in determining developmental patterning. It has been estimated that the orientation-related error could be as high as 20–50% of the total measurement error for Bcd gradients (*Gregor et al., 2007a*) and gap genes (*Dubuis et al., 2013*). This is because the dynamic developmental profiles vary spatially in the asymmetric 3D embryos, yet traditionally they are measured in a selected plane of the manually oriented embryo (*Gregor et al., 2007a*; *Houchmandzadeh et al., 2002*).

Here, we developed measurement methods to quantify the dynamic 3D expression of patterning genes in *Drosophila* embryos and applied these methods to measure the positional errors of the segmentation genes at different levels. We focused on one *stau⁻* allele, *stau^HL54*, which is reported to induce the largest variability in $x_{Hb}$ location (*Houchmandzadeh et al., 2002*). Surprisingly, we discovered that in *stau⁻* mutants, $x_{Hb}$ moves posteriorly by more than 10% EL to the WT position in nc14, and that the variability of $x_{Hb}$ location is comparable with that in the WT. Moreover, the upstream Bcd gradients show gradient intensity noise equivalent to those in the WT from nc12 to nc14, and the downstream Even-skipped (Eve) and cephalic furrow (CF) show the same positional errors as those in the WT. We also constructed a minimal model and revealed that the extremely large shift of the Hb boundary in *stau⁻* mutants originates from both the flattened maternal Hb profiles and the altered Bcd gradients, with a ~ 65% decrease in amplitude and a 17% increase in length constant in *stau⁻* mutants compared with the WT. Our results suggest that the threshold-dependent model could be valid at early nc14, but that the gene network then adjusts, and the filtering mechanism is implemented at least from the maternal Bcd gradient to Hb. Some factors other than *stau*, which remain to be discovered, could play important roles in filtering dynamic positional information.

## Results

### A reproducible large dynamic shift in the Hb boundary of *stau⁻* mutants

We observed a large dynamic shift of $x_{Hb}$ in *stau⁻* mutants using quantitative imaging with carefully controlled measurement errors. To control the measurement errors, we quantified the Hb expression pattern using 3D imaging with light-sheet microscopy (for more details, see 'Materials and methods', *Figure 1—figure supplement 1*, and *Figure 1—videos 1* and *2*) on fixed and immunostained embryos and staged each embryo with ~1 min temporal precision using the depth of the furrow canal (FC) (*Figure 1—figure supplement 2*). The average temporal Hb expression profiles of the WT (*Figure 1—figure supplement 3*) and of *stau⁻* mutants (*Figure 1A*) are shown on heat maps projected from 3D embryo images. From the heat map, we extracted the average of the normalized dorsal profile of Hb with a time-step of 5 min in nc14 (*Figure 1B and C*) and measured $x_{Hb}$. Consistent with previous results (*Dubuis et al., 2013*), $x_{Hb}$ of the WT remains nearly constant 20 min into nc14. Notably, in the first 20 min of nc14, it moves posteriorly by 3% EL (*Figure 1D*). By contrast, at the dorsal side of the embryo, $x_{Hb}$ of *stau⁻* mutants starts at 37.6 ± 1.9% EL at 2.5 min into nc14, then dynamically moves posteriorly by 10.3% EL within 60 min, and finally stabilizes at 47.9 ± 0.9% EL, close to the WT boundary position 47.8 ± 0.9% EL at 42.5 min into nc14 (*Figure 1D* and *Figure 1—figure supplement 4C and E*). Besides *stau^HL54*, another *stau⁻* allele, *stau^D3*, also shows a

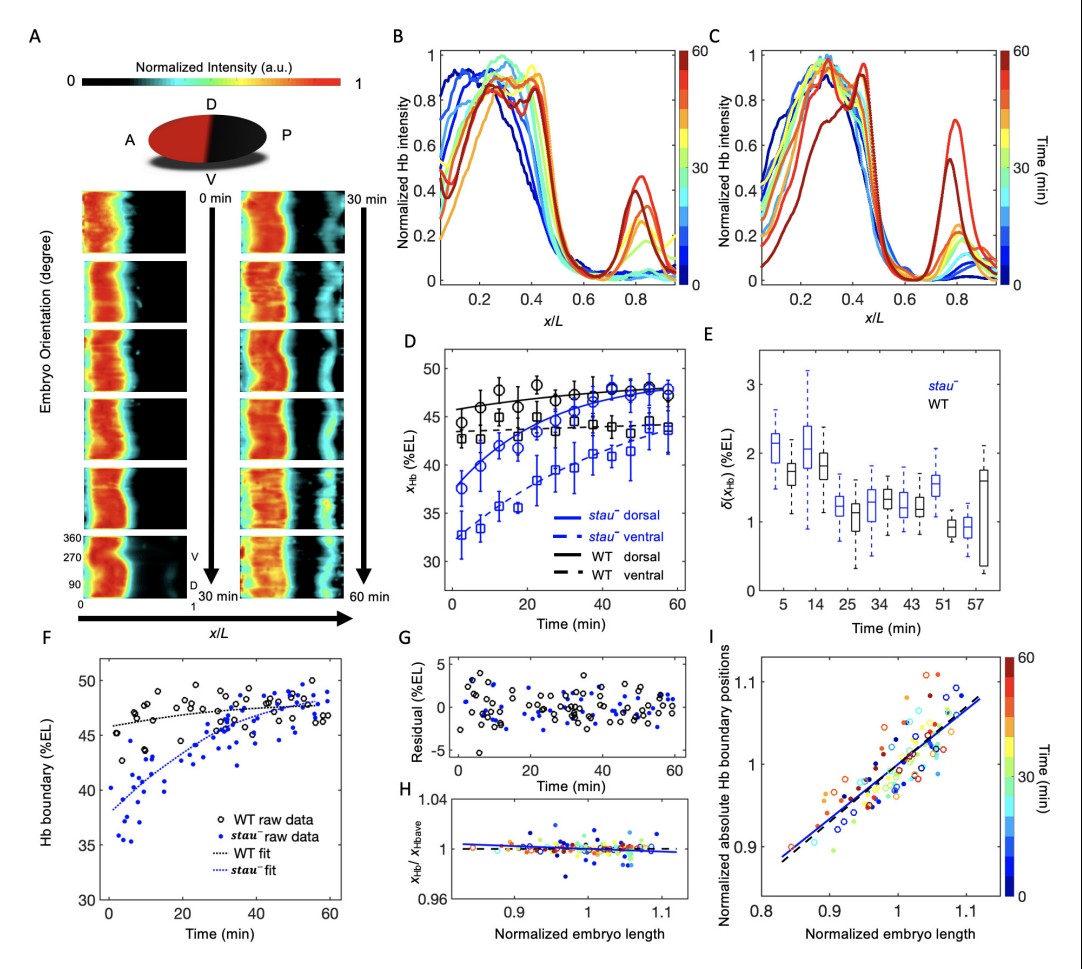

**Figure 1.** The Hb boundary reproducibly moves posteriorly in a much larger range in *stau⁻* mutants than in the WT based on 3D imaging. (**A**) Heat map of the normalized Hb intensity in *stau⁻* mutants with a time-step of 5 min in nc14 based on projection at different angles along the AP axis from the 3D imaging on embryos. (**B, C**) Dynamics of the average dorsal Hb profiles extracted from 3D imaging of the embryos of *stau⁻* mutants (**B**) and the WT (**C**) in nc14. (**D**) The average Hb boundary position ($x_{Hb}$) in each 5 min bin of *stau⁻* mutants (blue) and the WT (black) as a function of the developmental time in nc14. Each circle (square) represents the average of the dorsal (ventral) boundary positions of Hb in the bin, and error bars denote the standard deviation of the bin. Lines are eye guides. (**E**) The variability of the Hb boundary in each bin with equal sample numbers of *stau⁻* mutants (blue, $n = 10$ in each bin except the last one) and the WT (black, $n = 7$ in each bin except the last one) as a function of the developmental time in nc14. Error bars are calculated from bootstrapping. (**F, G**) De-trend analysis of the Hb boundary. When we used smooth splines to fit the Hb boundary of all the measured embryos as a function of the embryo development time into nc14 (**F**), the fitting residuals (**G**) for the *stau⁻* mutants (blue) and the WT (black) are comparable with each other. The standard deviations of the residual are 1.67% EL ($n = 69$) and 1.45% EL ($n = 47$) for *stau⁻* mutants and the WT, respectively. Two-sample *F*-test for equal variances cannot be rejected (p=0.31). (**H, I**) Scaling analysis of the Hb boundary. The plots show normalized relative Hb boundary position (**H**) or normalized absolute Hb boundary positions with respect to the anterior pole (**I**) versus normalized embryo length for *stau⁻* mutants (solid circles, solid line) and the WT (open circles, dashed line). The slopes of the linear regression lines fitting to all the data points for the WT and *stau⁻* mutants are 0.0004 ($R^2 = 0.0003$, $n = 47$) and −0.02 ($R^2 = 0.028$, $n = 69$), respectively (**H**), or 0.70 ($R^2 = 0.69$, $n = 47$) and 0.66 ($R^2 = 0.66$, $n = 69$), respectively (**I**). Each embryo length is normalized by the average embryo length of all embryos. The Hb boundary position is normalized by the corresponding fitting value in the de-trend analysis (**F**).

The online version of this article includes the following video, source data, and figure supplement(s) for figure 1:

**Source data 1.** Average Hb profiles and single embryo boundary positions.
**Source data 2.** The dynamics of the average and variability of Hb boundaries with equal sample size binning.
**Figure supplement 1.** 3D imaging measurement procedure.
**Figure supplement 2.** Age determination of fixed *Drosophila* embryos.
**Figure supplement 3.** Heat map of the Hb profiles of the WT taken from the projection.
**Figure supplement 4.** Dynamics of Hb profiles in *stau^{HL54}* mutants.
**Figure supplement 5.** Dynamics of Hb profiles in *stau^{D3}* mutants.
**Figure supplement 6.** Dependence of $x_{Hb}$ variability on spatial and temporal measurement errors.
*Figure 1 continued on next page*

Figure 1 continued
**Figure 1—video 1.** The z-stack of the immunofluorescence images of DAPI of the representative *stau⁻* mutant embryo after fusion in 3D imaging.
https://elifesciences.org/articles/54276#fig1video1
**Figure 1—video 2.** The z-stack of the immunofluorescence images of Hb of the representative *stau⁻* mutant embryo after fusion in 3D imaging.
https://elifesciences.org/articles/54276#fig1video2

large shift of $x_{Hb}$: 42.8 ± 1.5% EL to 47.9 ± 1.8% EL from 12.5 min to 52.5 min into nc14 on the dorsal side (*Figure 1—figure supplement 5*). It is well known that *stau^{D3}* is a strong allele with a fully penetrant abdominal segmentation phenotype (*Lehmann and Nüsslein-Volhard, 1991*), hence the observed large shift of $x_{Hb}$ in *stau⁻* mutants is not an allelic specific feature for *stau^{HL54}*, which could be hypomorphic, that is with partial function loss.

Moreover, $x_{Hb}$ differs significantly in different orientations, for example, the ventral boundary moves from 32.7 ± 2.4% EL by 10.9% EL to 43.6 ± 1.2% EL, with an anterior shift of approximately 6% EL compared with the dorsal boundary (*Figure 1D* and *Figure 1—figure supplement 4A,D and F*). By contrast, the difference of $x_{Hb}$ between the dorsal and ventral sides in the WT is only 3% EL (*Figure 1D*). These results suggest that it is crucial to control spatial and temporal measurement errors in order to assess the variability of the Hb boundaries. Interestingly, the spatial measurement error of $x_{Hb}$ seems to be the smallest on the dorsal profiles (*Figure 1—figure supplement 6A and B*).

Although $x_{Hb}$ dynamically moves posteriorly by more than 10% EL in *stau⁻* mutants, this shift is reproducible from embryo to embryo. Within a short time window, the variability of $x_{Hb}$ in *stau⁻* mutants is comparable with that of the WT (*Figure 1E*). Furthermore, even for all of the data from the whole nc14, after de-trending the dynamic shift, the standard deviations of $x_{Hb}$ for *stau⁻* mutants is 1.67 ± 0.16% EL (errors are estimated with bootstrap), similar to 1.45 ± 0.14% EL for the WT. Moreover, the variance difference between the two fly lines is not statistically significant (p=0.31 in a two-sample *F*-test for equal variances) (*Figure 1F–G*). These results seem to contradict previous measurements, which suggest that the variability of $x_{Hb}$ in *stau⁻* mutants is significantly higher than that of the WT (*Houchmandzadeh et al., 2002*; *He et al., 2008*). We suspect that the apparent difference in variability of $x_{Hb}$ might result from the different control of the measurement errors (*Figure 1—figure supplement 6C and D*). For instance, the temporal control in He's measurements (*He et al., 2008*) could be better than Houchmandzadeh's measurements (*Houchmandzadeh et al., 2002*), and the spatial control could be better in our measurements than in He's measurements (*He et al., 2008*). In *stau⁻* mutants, $x_{Hb}$ significantly varies as the development time or embryo orientation changes, so the measured variability of $x_{Hb}$ is very sensitive to spatial or temporal measurement errors. For example, the dynamic shift of $x_{Hb}$ has a strong effect on the calculation of the variability of $x_{Hb}$. If we pool all of the measured $x_{Hb}$ on the dorsal side, the standard deviation of $x_{Hb}$ is 3.5 ± 0.3% EL for *stau⁻* mutants, much greater than the WT (1.7 ± 0.2% EL). Moreover, the large shift of $x_{Hb}$ from the dorsal to the ventral side also strongly affects the measured variability of $x_{Hb}$. If we pool all of the measured $x_{Hb}$ values on both the dorsal and ventral side, the standard deviation of $x_{Hb}$ for *stau⁻* mutants can even increase to 4.5 ± 0.3% EL (*Figure 1—figure supplement 6C*).

The variability of $x_{Hb}$ also depends on whether it is scaling with the embryo length (*Houchmandzadeh et al., 2002*; *He et al., 2008*). Consistent with the comparable variability, *stau⁻* mutants and the WT are also similar in the scaling of the Hb boundary. On the one hand, the normalized relative Hb boundary positions are nearly constant for both *stau⁻* mutants and the WT (*Figure 1H*). On the other hand, the normalized absolute Hb boundary positions with respect to the anterior pole are proportional to the embryo length ($R^2$ = 0.69 and 0.66 for the WT and *stau⁻* mutants, respectively, *Figure 1I*). Notably, the scaling of $x_{Hb}$ of *stau⁻* mutants is significant only if we de-trend the dynamic shift, that is normalize the absolute Hb boundary position of each embryo by the average position in the corresponding developmental time. Without this normalization, the scaling of the absolute Hb boundary position of *stau⁻* mutants ($R^2$ = 0.26) is inferior compared with the WT ($R^2$ = 0.71), consistent with previous results (*He et al., 2008*).

## Bcd gradient noise of *stau⁻* mutants is nearly the same as that of the WT

To understand the origin of the variability of the Hb boundary, we measured its upstream Bcd gradients. Using live imaging with a two-photon microscope (TPM), we found that the average Bcd-GFP gradient of *stau⁻* mutants at 16 min into nc14 shows reduced amplitude but an increased length constant (*Figure 2A*). Compared with the WT, in *stau⁻* mutants the amplitude of the Bcd-GFP gradient is only 48 ± 7%, but the length constant is increased by approximately 13% from 18.6% EL to 21.1% EL (*Figure 2A*). If we take the GFP maturation effect into consideration (*Little et al., 2011*; *Liu et al., 2013*), the difference of the length constants can be even greater, increasing by 17% from 15.5% EL of the WT to 18.2% EL in *stau⁻* mutants (*Figure 3—figure supplement 1D*). This apparent increase of the measured length constant could result from more extensively distributed *bcd* mRNA in the embryos of *stau⁻* mutants (*Ferrandon et al., 1994*; *Petkova et al., 2014*), consistent with the simulation based on the synthesis-diffusion-degradation (SDD) model (*Grimm et al., 2010*; *Figure 3—figure supplement 1A–B*). Interestingly, the Bcd gradient noise of *stau⁻* mutants is

**Figure 2.** The Bcd gradient noise of *stau⁻* mutants is comparable with that of the WT. (A) The average intensity of the nuclear Bcd-GFP gradient of *stau⁻* mutants (blue) and the WT (black) as a function of the fractional embryo length at 16 min into nc14. Each circle and error bar represents the average and standard deviation, respectively, of the Bcd-GFP fluorescence intensity of the nuclei in the bin with a bin size of 2% EL. The inset shows the logarithm of the intensity as a function of the fractional embryo length and the linear fit in the range of 0.1–0.8 EL. (B) The relative Bcd-GFP gradient noise of *stau⁻* mutants (blue) and the WT (black). Each circle represents the standard deviation divided by the mean of the Bcd-GFP fluorescence intensity of the nuclei in the bin with a bin size of 2% EL. The error bar represents the standard deviation of the mean relative gradient noise calculated with bootstrap. (C) Variation in the average profiles of the Bcd gradient in *stau⁻* mutants from nc12 to 50 min into nc14. (D, E) The heat map of the Bcd gradient noise of the WT (D) and *stau⁻* mutants (E) from nc12 to 50 min into nc14. (F) The average Bcd gradient noise of *stau⁻* mutants from x = 0 to x = 0.55 EL versus developmental time is comparable with that of the WT.

The online version of this article includes the following source data and figure supplement(s) for figure 2:

**Source data 1.** Bcd gradients measured at 16 min into nc14.
**Figure supplement 1.** Live imaging of dynamic Bcd gradients.

comparable with that of the WT (*Figure 2B*). The relative noise of the Bcd-GFP gradient in the range of 0.1~0.6 EL is 18.0 ± 2.0%, which is very close to 17.9 ± 1.8% of the WT. Hence, it seems to be consistent with the comparable variability of the Hb boundary. However, these results are different from previous measurements (*He et al., 2008*; *Houchmandzadeh et al., 2002*), which might result from different measurement errors.

Another concern is that Bcd-dependent *hb* transcription turns off less than 10 min into nc14 (*Liu and Ma, 2013*; *Liu et al., 2016*), and that the time window for Bcd interpretation could be earlier than nc14 (*Huang et al., 2017*; *Bergmann et al., 2007*); hence Hb expression could be largely produced by translating the previously produced mRNA. It is therefore necessary to measure the dynamic Bcd gradient noise in developmental time earlier than the conventional measurement time, that is 16 min into nc14. By improving the imaging technique and image analysis method (*Figure 2— figure supplement 1*), we assessed the Bcd-GFP gradient noise in both *stau⁻* mutants and the WT. The average nuclear Bcd-GFP gradient rises to a maximum in nc14 and then slightly falls off

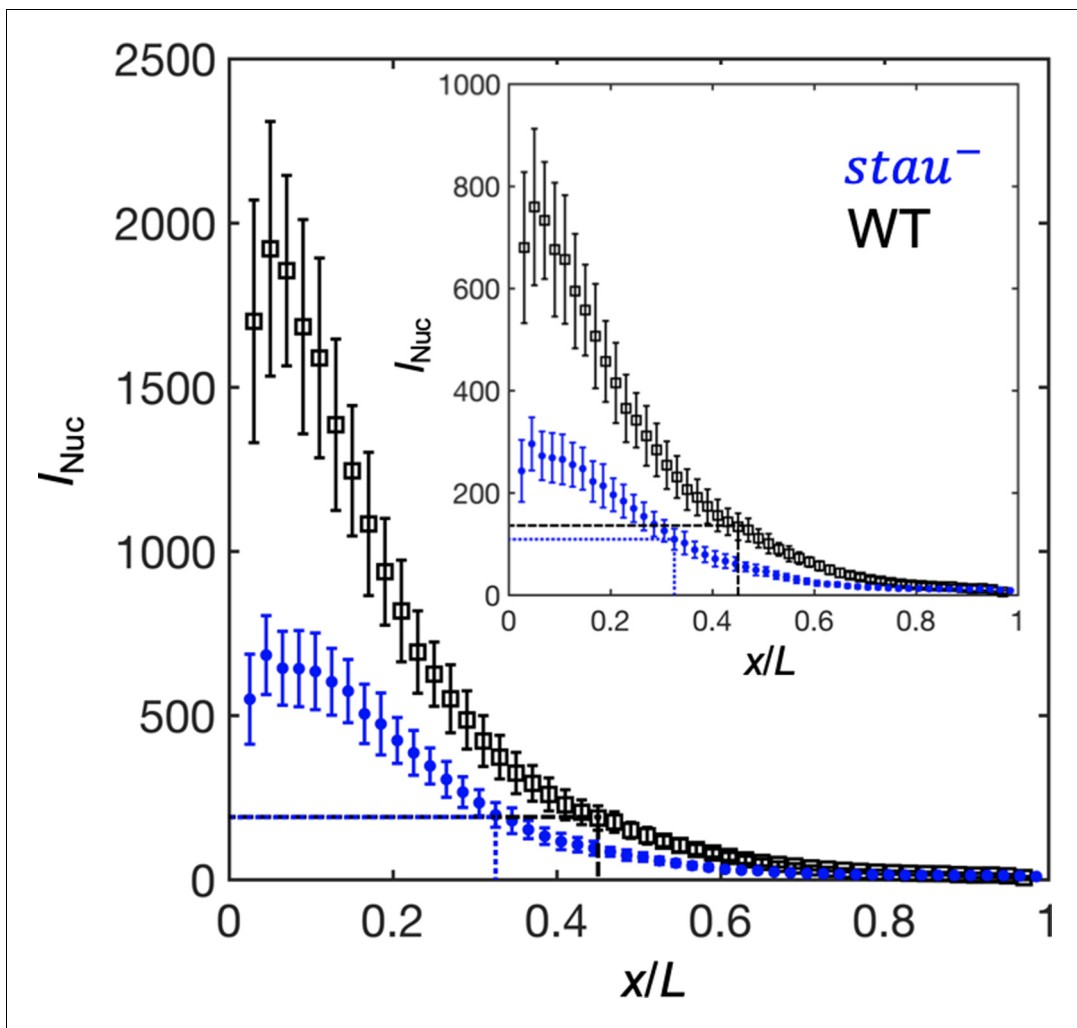

**Figure 3.** The activation of Hb by Bcd is consistent with the threshold-dependent model in *stau⁻* mutants at early nc14. Comparison of the Bcd concentration (horizontal lines) at $x_{Hb}$ (vertical lines) at 16 min into nc14 between *stau⁻* mutants (blue) and the WT (black) before (inset) and after maturation correction on the Bcd gradient. Each circle and error bar represent the average and standard deviation, respectively, of the Bcd-GFP fluorescence intensity of the nuclei in the bin with a bin size of 2% EL.

The online version of this article includes the following source data and figure supplement(s) for figure 3:

**Source data 1.** Source data for *Figure 3*.

**Figure supplement 1.** Maturation correction on Bcd-GFP gradients.

(*Figure 2C*). This is consistent with the results of another live imaging experiment (*Gregor et al., 2007b*) but is significantly different from the measurement of the fixed embryos (*Little et al., 2011*) because of the GFP maturation effect (*Little et al., 2011*; *Liu et al., 2013*). Importantly, the gradient intensity noise of *stau⁻* mutants and the WT remains comparable from nc12 to 60 min into nc14 (*Figure 2D–F*). It is also interesting to discover that the Bcd-GFP gradient noise in both fly lines remains at nearly the same level from nc13 to nc14 (*Figure 2F*). Notably, the measured gradient noise at 16 min into nc14 is slightly higher than that measured with TPM (*Figure 2B*), as the measurement error increases in the dynamic Bcd gradient measurements (e.g., a lower signal to noise ratio is due to the fast imaging speed and higher background). The measurement error, however, should change at a similar level for both *stau⁻* mutants and the WT, and hence their gradient noise is still comparable.

## The threshold-dependent activation model

As in *stau⁻* mutants, the amplitude of the Bcd-GFP gradient in the fly line with only half of the Bcd dosage (Bcd1.0) decreased by half compared with the WT (*Liu et al., 2013*). According to a previous study, for the fly line Bcd1.0, the Hb boundary position might be determined according to the threshold-dependent activation model in early nc14. Then, it dynamically moves posteriorly toward the Hb position of the WT but stops midway in later nc14 (*Liu et al., 2013*). By contrast, the Hb position in *stau⁻* mutants moves all the way to the WT position. It is interesting to investigate whether the threshold-dependent activation model is also valid in early nc14 in *stau⁻* mutants.

To test this speculation, we first needed to correct the GFP maturation effect to obtain the total Bcd-GFP gradient from the observed Bcd-GFP gradient measured with live imaging, as it takes tens of minutes for the newly synthesized Bcd-GFP to fluoresce. On the basis of the SDD model incorporating maturation correction and the spatial distribution of *bcd* mRNA (*Little et al., 2011*; *Petkova et al., 2014*), we calculated the maturation correction curves for the Bcd-GFP gradient in both *stau⁻* mutants and the WT (*Figure 3—figure supplement 1A*). Interestingly, the Bcd-GFP concentrations at the respective Hb boundary position are almost the same at 16 min into nc14 (*Figure 3*), suggesting that the threshold-dependent model might still apply in early nc14. However, as the $x_{hb}$ of *stau⁻* mutants moves posteriorly in a much greater range than that of the WT, and as the Bcd gradient changes only slightly in both fly lines, the exact time window in which this model applies remains to be determined.

## Positional information transmission

In addition to examining the threshold-dependent model, it was also interesting to test whether the noise-filtering model plays a role in early *Drosophila* embryogenesis. We measured the positional information transmission of the other patterning genes interacting with Hb in *stau⁻* mutants. As another gap protein, Kr, forms a strip adjacent to the Hb boundary, and the two genes suppress each other (*Jaeger, 2011*). In *stau⁻* mutants, the Kr strip shows a width of 29% EL, wider than that in the WT (17% EL) at the beginning of embryogenesis, then it narrows to 20% EL (*Figure 4—figure supplement 1A*). It then moves posteriorly together with the Hb boundary in early nc14 but ends its movement late in nc14 (*Figure 4A*), suggesting that the late shift in the Hb boundary is independent of Kr. As one of the downstream pair rule proteins, Eve forms only four instead of seven strips in *stau⁻* mutants (*Figure 4—figure supplement 1B*). The dynamic shift in the four strips is also shown in *Figure 4B*. At 56 min into nc14, the cephalic furrow (CF) position in the *stau⁻* mutants is 28.2 ± 1.1% EL, while the CF of the WT is 34.3 ± 1.4% EL.

To understand the dynamic transmission of positional information, we compared the positional noise of all the measured AP patterning markers from Bcd to CF during the time course of nc14 (*Figure 5*). Consistent with previous results, the positional error of the Hb boundary of the WT decreases by approximately two-fold to a minimum at approximately 45 min into nc14 and then slightly increases (*Dubuis et al., 2013*). The *stau⁻* mutants show a trend that is similar to that in the WT. The positional error of $x_{Hb}$ in early nc14 is slightly less than that of the Bcd gradient for *stau⁻* mutants and the WT. The minimal positional noise of the Kr front boundary, the first peak of Eve, and CF are comparable with the minimal positional error of the Hb, that is, approximately 1% EL. Notably, the *stau⁻* mutants and the WT are comparable in their minimal positional errors in these different levels

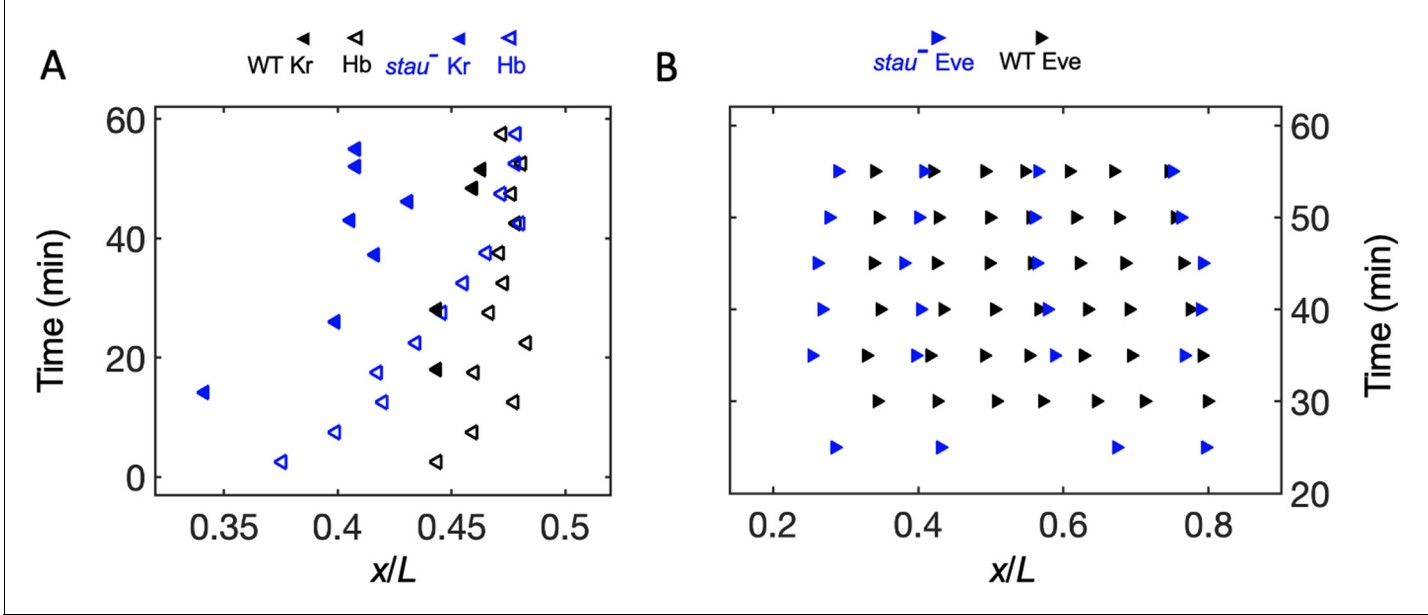

**Figure 4.** The dynamic shift range of the position of Kr and Eve is smaller than that of Hb in *stau⁻* mutants. (A) The positions of the anterior boundary (inflection point) of Kr (solid triangle) and $x_{Hb}$ (hollow triangle) with the *stau⁻* mutants (blue) and the WT (black). (B) The peak of Eve in *stau⁻* mutants (blue) as a function of the developmental time in nc14 in comparison with that in the WT (black).
The online version of this article includes the following source data and figure supplement(s) for figure 4:

**Source data 1.** Source data for *Figure 4*.
**Figure supplement 1.** Measurement results of Kr and Eve in *stau⁻* mutants.
**Figure supplement 2.** Representative raw images of Bcd-GFP, Hb, Kr, Eve, and CF of the WT and *stau⁻* mutants.
**Figure supplement 3.** Cuticle patterns of *stau⁻* mutants.

of AP patterning genes (*Figure 5*). These results suggest that positional noise is filtered from Bcd to Hb and is then relayed between different levels of the patterning genes.

In addition to the noise-filtering phenomenon, it is interesting to note that the shifts in the patterning markers differ. The Hb boundary moves posteriorly by more than 10% EL to nearly the WT position. For the other gap genes that interact with Hb, the front boundary of Kr moves posteriorly by only 5% EL and stops 3% EL away from the WT position. For the downstream genes, the first peak of Eve moves very little, and its final position (corresponding to the CF position) is 6% EL away from the WT position. These results indicate that the large dynamic shift of Hb seems to be dampened instead of faithfully relayed in the other patterning genes. By contrast, the shift accumulates in the fly lines that have altered Bcd dosages (*Liu et al., 2013*). For example, in the fly line Bcd1.0, which has almost the same Bcd dosage as *stau⁻* mutants, the Hb boundary position moves posteriorly by approximately 2.5% EL in nc14 and stops at 7% EL away from the WT position. The first peak of Eve moves posteriorly further by 0.8% EL, so the CF position is closer to the WT position. It has been suggested that this additive shift results from the dynamic integration of maternal positional information (*Liu et al., 2013*). It remains to be investigated whether a new mechanism may account for the dynamic positional transmission in *stau⁻* mutants.

## Discussion

### Improving quantitative measurement methods to reveal the true biological noise of developmental patterning

Maternal mutants become more accessible with the CRISPR-Cas9 technique (*Bassett and Liu, 2014*). The balancer stabilizes the maternal mutant fly line but prevents the random mutation from being rescued by homologous recombination. Hence, the accumulated mutations often deteriorate the maternal mutant fly line in the fly stock. In this work, instead of running rescue genetics

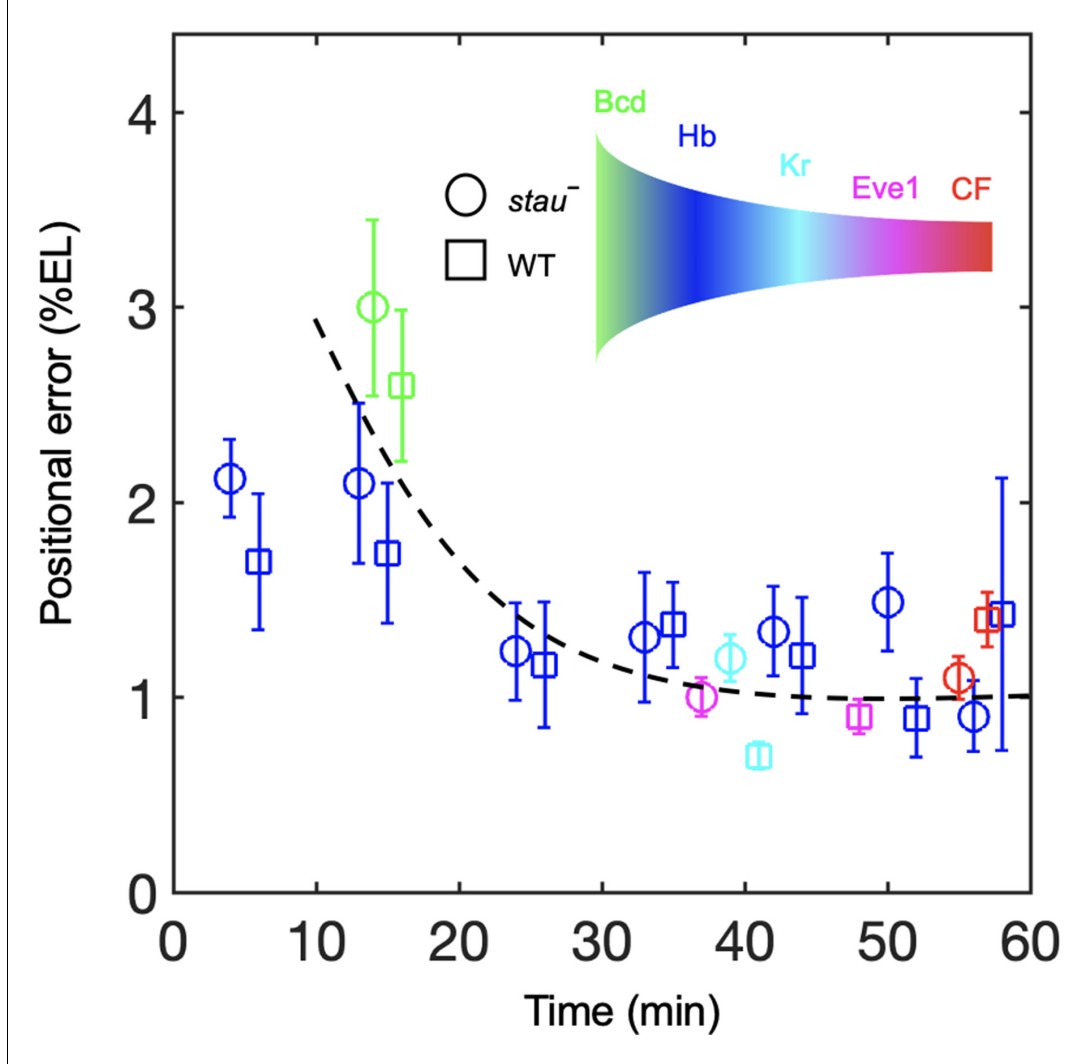

**Figure 5.** Positional noise is filtered from Bcd to CF. Positional errors in Bcd (green), Hb (blue), Kr (cyan), Eve (magenta) and CF (red) of *stau⁻* mutants (circle) and the WT (square) as a function of developmental time in nc14. Positional errors are calculated from the Bcd gradients (*Figure 2A*) after subtracting the imaging noise and image mask noise from the gradient noise (*Figure 2B*). The standard deviation of the average positional noise is calculated on the basis of bootstrapping. For presentation purposes, data for Bcd, Kr and CF measured at the same time point are shown with 1 min offset on the x axis. The black line connects the data points to guide the eye.

The online version of this article includes the following source data and figure supplement(s) for figure 5:

**Source data 1.** Comparison of the minimal positional errors of patterning markers.

**Figure supplement 1.** Dynamics of the average Hb boundary slopes.

experiments, we generated a fresh *stau⁻* mutant fly line from the WT by using the CRISPR-Cas9 technique. The newly generated *stau^HL54* mutant fly line shows the correct expression patterns of Bcd, Hb, Kr, and Eve compared with the original *stau⁻* mutant fly line (*Figure 4—figure supplement 2*), and the cuticle patterns of *stau^HL54* and *stau^D3* are consistent with the published record (*Lehmann and Nüsslein-Volhard, 1991*; *Figure 4—figure supplement 3*).

To reveal the true biological noise, 3D imaging is very helpful in controlling spatial measurement errors. Compared with the two-photon microscopy used in previous studies for 3D imaging of embryos (*Fowlkes et al., 2008*), the light-sheet microscope should provide better-quality 3D reconstruction by combining images taken from two opposite directions with higher imaging speed (*Krzic et al., 2012*). However, if strong scattering or absorbing objects exist in samples, caution is needed to alleviate or properly correct the potential striper shadow artifacts (*Mayer et al., 2018*). The conventional 2D expression profiles can be extracted with the 3D imaging analysis tools

developed by us and the others (*Fowlkes et al., 2008*; *Heemskerk and Streichan, 2015*). Using the heat map projected from 3D imaging data, we can conveniently evaluate the measurement errors in embryo orientations. Interestingly, most of the time, the measurement errors on the dorsal side are smaller than those on the ventral side and the symmetric two sides on the coronal plane (*Figure 1—figure supplement 6A and B*). This is because the Hb boundary moves in a smaller range around the dorsal side (*Figure 1A* and *Figure 1—figure supplement 3*). Nevertheless, even with 10° uncertainty in the orientation, the measurement errors on the dorsal side could still be as high as 0.2% EL, at least 20% of the positional error of the Hb boundary, which is 1% EL.

For dynamically evolved patterning, it is also important to determine the embryo age precisely. Both the depth of the FC and nuclear shape/size are good measures of the embryo age in nc14. The former has better time resolution in the late developmental stage. Notably, the dynamic shift in FC could vary in different fly lines, as the shifting curve measured with $w^{1118}$ is different from that described by previously published results measured with Oregon-R (*Figure 1—figure supplement 2A*; *Dubuis et al., 2013*).

When compared with the imaging of fixed embryos, live imaging has advantages in potentially higher temporal resolution determined by the imaging speed. Only in nc14, the densely packed nuclei on the embryo surface can be imaged in one plane. Hence, for convenience in controlling the measured error, the Bcd gradient noise is often measured only in a selected plane at a timepoint of 16 min into nc14. However, before this measurement time, the Bcd-dependent regulation of Hb has already been shut off (*Liu and Ma, 2013*). Moreover, considering the dynamics of downstream patterning genes, it is important to measure the dynamics of the Bcd gradient noise. We found that the maximum projection from a five-layer and 1-μm-spaced z-stack image could significantly alleviate the measurement errors. To accumulate sufficient samples to measure the gradient noise, it is also very important to stabilize the imaging condition and to correct the potential intensity drift between different experimental sessions (*Figure 2—figure supplement 1C and D*). With this imaging improvement as well as automatic image analysis, we successfully showed that the Bcd-GFP gradient noise is almost constant in the interphase from nc13 to nc14 (*Figure 2F*). As the fluorescence intensity observed in live imaging increases during this period, this result may suggest that the Bcd gradient noise is not dominated by the Poissonian noise.

## Underlying mechanism for the dynamic shift of the Hb boundary

On the basis of the quantitative spatial-temporal gene expression data, it has long been known that the gap gene profiles, for example, the central Kr domain and the posterior Kni and Gt domains, show substantial anterior shifts during nc14 (*Jaeger et al., 2004*). These dynamic shifts are proposed to originate from the asymmetric cross-regulation between gap genes, that is, posterior gap genes repress their adjacent anterior gap genes but not vice versa (*Huang et al., 2017*; *Jaeger et al., 2004*). The shift amount is less than 5%, much less than the shift of the Hb boundary in *stau⁻* mutants.

The Hb boundary and the other patterning features, such as the Kr central strip and Eve peaks, also move dynamically with differing Bcd dosages (*Liu et al., 2013*). For example, for the fly line with only half of the Bcd dosage in the WT, the shift in the CF is only approximately 40% of the predicted value based on the threshold-dependent model. The Hb boundary moves posteriorly toward the WT position by approximately 4% EL from an initial position close to the one predicted by the threshold-dependent model. The posterior shift of the Hb boundary has also been observed in *nos⁻* mutants (*Petkova et al., 2019*).

Among all the reported shifts of the patterning markers in fly embryos, the shift in the Hb boundary in *stau⁻* mutants is the largest at more than 10% EL. The shift significantly slows down in late nc14, as the majority of the shift, ~7% EL, finishes in the first 30 mins of nc14. By contrast, most of the shifts observed in the other cases occur in the later 30 mins of nc14 and have been proposed to originate from the cross-regulation between gap genes (*Jaeger et al., 2004*; *Liu et al., 2013*). Hence, a mechanism other than the cross-regulation between gap genes could contribute to the large shift of the Hb boundary in *stau⁻* mutants. Moreover, the slope of the Hb boundary in *stau⁻* mutants is less than that of the WT in the first 30 mins in nc14 and later increases to nearly the same as that of the WT (*Figure 5—figure supplement 1*). This might be related to the modification of the maternal gradients from both poles (*Figure 6A and B*).

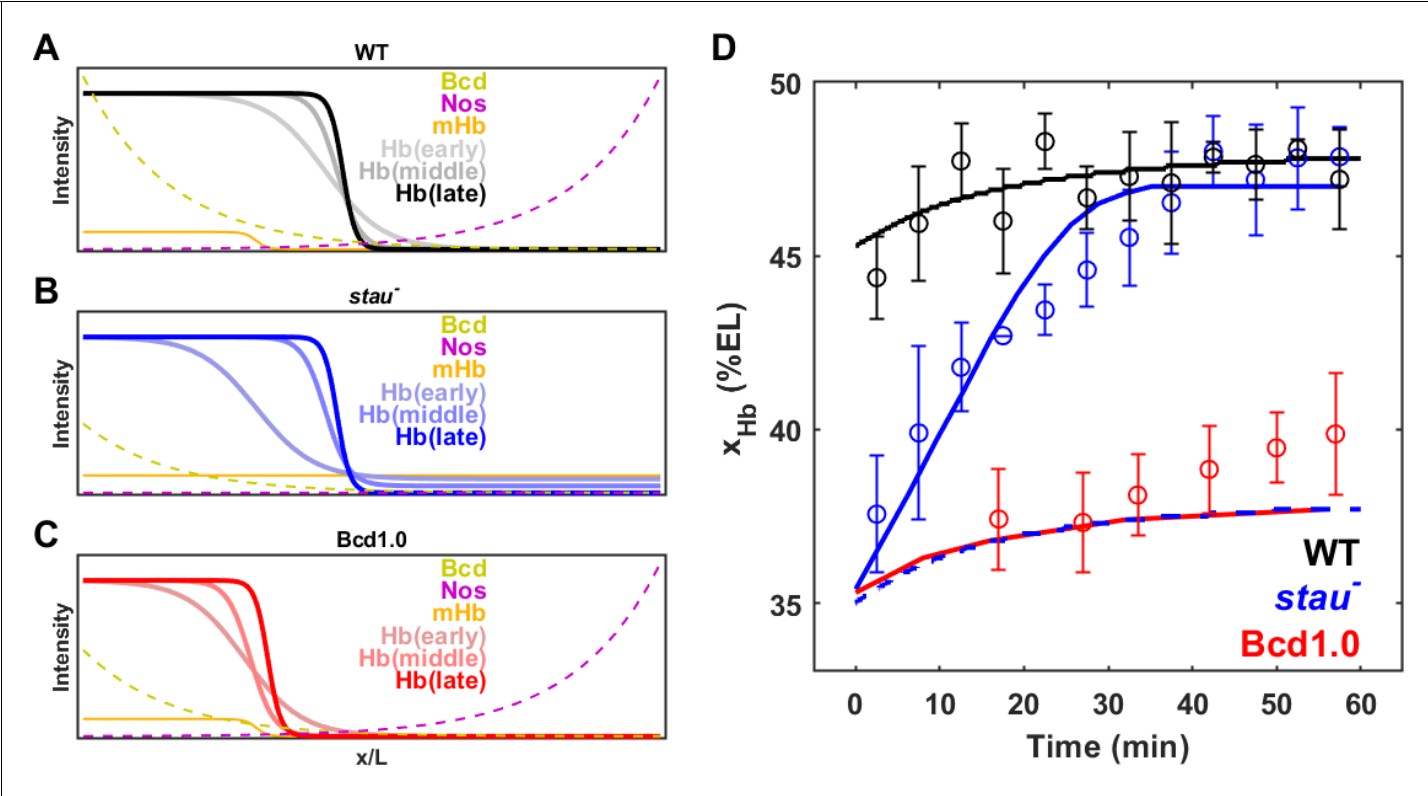

**Figure 6.** Origin of the large shift in $x_{Hb}$ in *stau*⁻ mutants. (A–C) Schematic of maternal gradients and Hb dynamic expression in the WT (A), *stau*⁻ mutants (B), and Bcd1.0 (C). In *stau*⁻ mutants, the Bcd gradient (yellow) decreases in amplitude and increases in length constant. The depletion of the Nos gradient (purple) flattens the maternal Hb gradient (orange). In Bcd1.0, the Bcd gradient decreases in amplitude by half. $x_{Hb}$ moves posteriorly from early, to middle and late nc14. It mainly moves in the first 30 mins in nc14 in *stau*⁻ mutants and the shift amount is much greater than in the WT. By contrast, $x_{Hb}$ mainly moves in the later 30 mins of nc14 in Bcd1.0. (D) The mathematical model fitting (lines) agrees well with the measured shift of $x_{Hb}$ in the WT (black circles) and *stau*⁻ mutants (blue circles), but not Bcd1.0 (red circles). The simulation result without the distortion effect on the Nos gradient in *stau*⁻ mutants (blue dashed line) fails to replicate the measured results. Error bars represent the standard deviation of $x_{Hb}$ in each time window.

The online version of this article includes the following source data and figure supplement(s) for figure 6:

**Source data 1.** Model parameters.
**Source data 2.** Source data for *Figure 6* and *Figure 6—figure supplement 1*.
**Figure supplement 1.** Mathematical modeling of the shift of $x_{Hb}$.

To test this idea, we constructed a mathematical model to calculate the dynamic shift in $x_{Hb}$ in both *stau*⁻ mutants and the WT. To illustrate the Stau effect, we neglected the cross-regulation between different gap genes and only considered the activation from Bcd to Hb and the self-activation of Hb. We assume that *stau*⁻ mutants and the WT share the same gene regulation function for Hb, and that the only difference comes from the distorted expression profiles of Bcd and maternal Hb in *stau*⁻ mutants (for more details, see 'Materials and methods', *Figure 6A–C*, and *Figure 6—figure supplement 1A*). On the one hand, the amplitude of the Bcd gradient is reduced by 65%, and the length constant is increased by 17%. On the other hand, the maternal Hb profile is flattened as the Nos gradient is removed. This model fits well with the measured dynamic shift of $x_{Hb}$ in both *stau*⁻ mutants and the WT (*Figure 6D*), indicating that the synergy effect resulting from the altered Bcd gradients and maternal Hb in *stau*⁻ mutants can account for the much larger shift of $x_{Hb}$ compared with that in the WT.

The depletion of the Nos gradient by the *stau*⁻ mutant is necessary for the observed larger shift of $x_{Hb}$ in *stau*⁻ mutants. If only the amplitude of Bcd gradients is altered, this model predicts that the shift of $x_{Hb}$ is much smaller. If we rescue the Nos gradient, that is keep the maternal Hb in *stau*⁻ mutants the same as that in the WT, the simulation based on the fitted model shows that $x_{Hb}$ starts

at 35% EL, but only moves posteriorly by ~1% EL (*Figure 6D*). A very similar result is also observed in the simulated dynamics of $x_{Hb}$ in the fly line Bcd1.0, of which the Bcd dosage is only half of that in the WT. The shift amount of $x_{Hb}$ in Bcd1.0 is only 1%EL, smaller than the experimental value of 4% EL (*Figure 6D*). Although the initial position of $x_{Hb}$ might agree with the experiment, the simulated shift slows down significantly in later nc14, inconsistent with the unchanged shifting speed in the experiment. We speculate that the shift in late nc14 could be attributed to the cross-regulation between gap genes, as suggested in previous studies (*Liu et al., 2013*).

Moreover, this model predicts that the initial position of $x_{Hb}$ varies as the Bcd gradient is tuned. Our measurement shows that Bcd gradients on the ventral side decrease in the amplitude and increase in the length constant compared with those on the dorsal side (*Figure 6—figure supplement 1B*), consistent with previous results (*Gregor et al., 2007a*). This difference could be attributed to the observed anterior shift of $x_{Hb}$ on the ventral side, which is approximately 6% EL for *stau⁻* mutants but 3% EL for the WT (*Figure 6—figure supplement 1C*). This model also suggests that the dynamic change of Bcd gradients (*Little et al., 2011*) influences the dynamic shift of $x_{Hb}$, as it predicts that $x_{Hb}$ could continue to move posteriorly (*Figure 6—figure supplement 1D and E*) if a static Bcd gradient is applied. This prediction is inconsistent with the experimental observation that *stau⁻* mutants and the WT show reduced shift speeds and stabilized $x_{Hb}$ at the end of nc14.

Hence, the current model suggests that the ultra-large shift of $x_{Hb}$ in *stau⁻* mutants can be attributed to the distorted Bcd and Nos gradients due to the loss of Stau function, if self-activated Hb is activated by Bcd according to the threshold-dependent model at early nc14. However, this simplified model without cross-regulation of the gap genes fails to predict the small shift of $x_{Hb}$ in *nos⁻* mutants (*Houchmandzadeh et al., 2002*; *Petkova et al., 2019*). Thus, a comprehensive model incorporating both the maternal factors and the gap genes is still needed to dissect the sophisticated role of Stau in regulating Hb patterning.

## Dynamic transmission of positional information

The large dynamic shift in the Hb boundary raises the question of how the positional information is transferred in the patterning system. The term 'positional information' was first coined by *Wolpert, 1969*. On the basis of this concept, developmental patterning is instructed by the concentration of a single static morphogen gradient, and the interpretation of the morphogen gradient follows the threshold-dependent model. Each cell 'acquires' its position inside the embryo by 'reading' the morphogen concentration and accordingly activates downstream genes to form the cell-fate map (*Jaeger and Reinitz, 2006*; *Wolpert, 2011*; *Wolpert, 2016*). This model provides a simple molecular-based mechanism for developmental pattern formation. It has prevailed in developmental biology, especially after the identification of a series of morphogens starting with Bcd (*Rogers and Schier, 2011*; *Struhl et al., 1989*).

However, this model has long been challenged because gradient noise could disrupt patterning precision (*Houchmandzadeh et al., 2002*; *Jaeger et al., 2007*). Without precise morphogen gradients as inputs, the developmental pattern could also form via a noise-filtering mechanism resulting from cross-regulation between genes (*Manu et al., 2009*). This idea is rooted in Turing's seminal idea: periodic patterns can spontaneously form in a self-organizing reaction-diffusion system, for example, a slow diffusive activator and a fast diffusive inhibitor (*Corson and Siggia, 2012*; *Turing, 1952*). Recently, an increasing number of developmental patterning systems have been found to implement a Turing-like mechanism (*Economou et al., 2012*; *Goryachev and Pokhilko, 2008*; *Raspopovic et al., 2014*).

Usually, these two classic mechanisms have been thought to be mutually exclusive. Hence, great effort has been made to distinguish which mechanism is implemented in a particular developmental system. However, whether early fly embryogenesis follows the threshold-dependent model (*Gregor et al., 2007a*; *He et al., 2008*) or the noise-filtering model (*Houchmandzadeh et al., 2002*; *Manu et al., 2009*) has been controversial because it has not been easy to run quantitative tests. First, it is well known that the gap protein profiles dynamically change in nc14 (*Dubuis et al., 2013*; *Jaeger et al., 2004*). Hence, the developmental system is not in a steady state, and the concentration of these transcription factors could continue to change with a time scale not much longer than the degradation time of the downstream gene. As a result, the developmental pattern cannot be regarded as the instant readout of simultaneous transcription factors but instead should be seen as

an accumulation of the product generated in an early time window. Second, the gap gene integrates multiple maternal factors (*Jaeger, 2011*; *Liu et al., 2013*). Hence, we need to measure the combined positional information of all the upstream genes and the dynamic positional information of the downstream genes. On the one hand, however, the conventional measurement method without sufficient control of spatial and temporal measurement error is rather limited in its ability to measure expression dynamics. On the other hand, the combined positional information is difficult to calculate because the regulatory function is still unknown, although it might be estimated on the basis of the optimal decoding hypothesis (*Petkova et al., 2019*).

By developing 3D measurements with reduced measurement errors, we observe that the positional errors decrease from approximately 2.0% EL at early nc14 to approximately 1.0% EL in the middle of nc14 in *stau⁻* mutants. A slight decrease in the positional errors was also observed in the WT (*Figure 5*). Interestingly, even for *stau⁻* mutants with an ultra-large dynamic shift in the Hb boundary, the threshold for Bcd in activating Hb still appears to be the same as that of the WT at 16 min into nc14. These results suggest that the threshold-dependent positional information model probably acts upstream of the self-organized filtering mechanism during *Drosophila* embryogenesis. This scenario is actually one of the most basic 'building blocks' of the interaction between the two mechanisms (*Green and Sharpe, 2015*). We expect that studying the combination of the two mechanisms could be the key to revealing the mechanism of precise and robust pattern formation via dynamic transmission of positional information. Our newly developed measurement methods, together with recent developed dynamic measurement (*Bothma et al., 2018*; *Dubuis et al., 2013*; *Durrieu et al., 2018*; *Garcia et al., 2013*; *Huang et al., 2017*; *Lucas et al., 2013*) and modeling (*Verd et al., 2017*) tools, will facilitate the characterization of the dynamic transmission of positional information.

# Materials and methods

## Key resources table

| Reagent type (species) or resource | Designation | Source or reference | Identifiers | Additional information |
|---|---|---|---|---|
| Strain, strain background (*D. melanogaster*) | $w^{1118}$ | Yi Rao Lab | BDSC3605 | |
| Strain, strain background (*D. melanogaster*) | $stau^{HL54}$ | This paper | | See 'Materials and methods' |
| Strain, strain background (*D. melanogaster*) | $stau^{D3}$ | This paper | | See 'Materials and methods' |
| Strain, strain background (*D. melanogaster*) | $bcd\text{-}egfp;+;bcd^{E1}$ | Thoms Gregor Lab | | |
| Strain, strain background (*D. melanogaster*) | $bcd\text{-}egfp;$ $stau^{HL54};bcd^{E1}$ | Thoms Gregor Lab | | |
| Antibody | Anti-Hb (mouse monoclonal) | Abcam | ab197787 | IF (1:1000) |
| Antibody | Anti-Kr (guinea pig polyclonal) | John Reinitz Lab | | IF (1:1000) |
| Antibody | Anti-Eve (rat polyclonal) | John Reinitz Lab | | IF (1:1000) |
| Antibody | Anti-guinea pig (goat monoclonal) | John Reinitz Lab | | IF (1:1000) |
| Antibody | Anti-rat (goat monoclonal) | Invitrogen | A11006 | IF (1:1000) |
| Antibody | Anti-mouse (goat monoclonal) | Invitrogen | A21240 | IF (1:1000) |

*Continued on next page*

*Continued*

| Reagent type (species) or resource | Designation | Source or reference | Identifiers | Additional information |
|---|---|---|---|---|
| Chemical compound, drug | Agarose gels | Invitrogen | E-Gel EX | 1~1.5% |
| Software, algorithm | 3D image processing | This paper | Source code files | See 'Materials and methods' |
| Software, algorithm | MATLAB | *MATLAB, 2018* | MATLAB 2018a v9.4.0.813654 | |
| Other | DAPI stain | Invitrogen | D1306 | (1 µg/mL) |
| Other | Capillary | Brand | 701904 | |

## Fly strains

Bcd-GFP intensity was measured in the fly strains *bcd-egfp;+;bcd$^{E1}$* and *bcd-egfp;stau$^{HL54}$;bcd$^{E1}$* using live imaging. The expression profiles of Hb, Kr, and Eve in immunostained embryos were measured with *w$^{1118}$*, *stau$^{HL54}$* (*w;stau$^{HL54}$*) and *stau$^{D3}$* (*w;stau$^{D3}$*) mutants. The two *stau$^-$* mutants were generated from *w$^{1118}$* directly by corresponding gene editing of the RNA-binding domain of *stau* using CRISPR/Cas9 (*Bassett and Liu, 2014*): a point mutation (replacing the base T in the intron of the fifth exon with base A) for *stau$^{HL54}$*, and a replacement of the second to fifth exon regions with a knock-in *rfp* marker for *stau$^{D3}$*.

## Immunostaining

All embryos were collected at 25°C, dechorionated, and then heat-fixed in 1x TSS (NaCl, Triton X-100). After at least 5 min, the embryos were transferred from the scintillation vial to Eppendorf tubes and vortexed in 1:1 heptane and methanol for 1 min to remove the vitelline membrane. They were then rinsed and stored in methanol at −20°C. The embryos were then stained with primary antibodies, including mouse anti-Hb (Abcam, ab197787), guinea pig anti-Kr, and rat anti-Eve (gifts from John Reinitz). Secondary antibodies were conjugated with Alexa-647 (Invitrogen, A21240), Alexa-488 (Invitrogen, A11006), and Alexa-555 (gift from John Reinitz). To prevent cross-binding between the rat and mouse primary antibodies, the embryos were incubated in first guinea pig and then rat primary antibodies, followed by their respective secondary antibodies. Subsequently, the embryos were treated in blocking buffer before the mouse primary and secondary antibodies were applied. Finally, the embryos were stained with DAPI (Invitrogen, D1306).

## 3D imaging and analysis

Embryos that were stained and washed together in the same tube were mounted in agarose (Invitrogen E-Gel EX Agarose gels, 1~1.5%) in a capillary tube with an inner diameter of 1 mm (Brand, 701904). Samples were imaged on a Zeiss Z1 light-sheet microscope. Images were taken with a W Plan Apo 20X/1.0 water immersion objective and with sequential excitation wavelengths of 638, 561, 488 and 405 nm. The thickness of the light sheet was 4.6 or 4.35 µm. Before the fluorescence imaging, each embryo was adjusted to the maximal sagittal plane via bright-field imaging. For each embryo, two z-stacks of images (1920 × 1920 pixels, with 16 bits and a pixel size of 286 nm at 0.8 magnified zoom) with 1 µm spacing were taken from two opposite sides by rotating the embryos 180° (*Figure 1—figure supplement 1A*). Several factors are key to improving image quality. i) Light sheet illumination setting: chose the 'Dual Side when Experiment' and the 'Online Dual Side Fusion', then adjusted the left and right beam path to get the optimal images. ii) Pivot scan setting: the 'Pivot scan' was activated to reduce the shadows that might otherwise be cast by optically dense structures within the embryos. iii) Laser power and exposure time: adjust laser power and exposure time to maintain an adequate signal to noise ratio and to avoid severe photo-bleaching.

Image analysis routines were implemented in customized MATLAB codes (*MATLAB, 2018*) with five steps. i) Image-registration and fusion. Two z-stacks of raw images of each embryo were acquired from opposite directions by rotating 180°. The two stacks were registered with the auto-correlation algorithm (*Figure 1—figure supplement 1B*), which is based on the correlation coefficient reflecting the similarity between two images. Next, the fused images were obtained by averaging the two raw images at the same position and opposite direction (*Figure 1—figure*

*supplement 1B*). This fusion process compensates the dependence of the intensity on the imaging depth (*Figure 1—figure supplement 1E*). ii) 3-D reconstruction. The 3D embryos were reconstructed with these fused z-stack images by the 3D interpolation algorithm (*Figure 1—figure supplement 1C*). iii) Segmentation at different angles. A plane across the AP axis with the largest area was identified. Then, starting from this plane, 36 sides of 2D images were extracted from the 3D embryos by rotating slices with 5° intervals at different orientations along the AP axis (*Figure 1—figure supplement 1C*). iv) Profile extraction. The gene expression profiles were extracted from these slices by morphology image processing to generate the embryo mask, extract the intensity of nuclei surrounding the boundary of embryos mask, and obtain the profile at different embryo orientations. v) Heat map construction. The 36 Hb intensity profiles of one embryo were normalized from 0 to 1, then placed in a heat map, in which the dorsal side was located in the bottom quarter and the ventral side was located in the upper quarter (*Figure 1—figure supplement 1D*).

## Live imaging and analysis

Embryos were collected at 25°C for 1 hr, dechorionated in 100% bleach (4% NaClO) for 3 min, glued on a glass slide and covered with halocarbon oil (Halocarbon 700). The Bcd gradient in the mid-coronal plane at 16 min into nc14 was measured as reported previously. In brief, embryos were imaged at 22°C on a TPM that was built in house. The excitation laser was 25 mW in average power and 970 nm in wavelength. The objective was a Zeiss 25X (NA = 0.8 in air) oil/water immersion objective. Emission fluorescence was collected with a gallium-arsenide-phosphide (GaAsP) with a quantum yield of more than 40% and dark counts of less than 4000/s at 25°C. For each embryo, three images (512*512 pixels with a pixel size of 460 nm, bit depth of 12 bits, and scan speed of 4 ms/line) were taken sequentially along the A-P axis and stitched together. In each session, embryos from the fly strains *bcd-egfp;+;bcd$^{E1}$* and *bcd-egfp;stau$^{HL54}$;bcd$^{E1}$* were mounted on the same slides and measured side by side.

For the dynamic Bcd gradient measurements, the embryos were glued on the cover glass of a Petri dish of 30 mm in diameter (NEST, 801002) and covered with halocarbon oil. They were imaged in the mid-sagittal plane at 25°C on a Nikon A1RSi+ confocal microscope with a Nikon Plan Apo λ 20X/0.75 air objective. The fluorescence was excited at a wavelength of 488 nm and collected with a GaAsP detector. A maximum-intensity-projected *z*-stack of five images (1024 × 1024 pixels with a pixel size of 620 nm, bit depth of 12 bit, and spacing of 1 μm) around the largest plane was acquired at each time point. Each of the images was averaged on two sequential acquisitions. In each session, at most three embryos were picked for imaging to guarantee the time resolution (80 s, scan speed of 2.3 s/frame). The background was the average of the background fluorescence measured with ten $w^{1118}$ embryos under the same conditions. To correct for the potential imaging differences in different sessions, the control samples were prepared with the hand-peeling protocol to preserve the fluorescence of the Bcd-GFP of the collected embryos. Embryos at the interphase of nc13–nc14 were selected and imaged in advance of each imaging session. Imaging analysis was processed with customized MATLAB codes (*MATLAB, 2018*).

## Model

Hb patterning is treated as a one-dimensional reaction-diffusion system with no-flux boundary condition:

$$\frac{\partial h(x,t)}{\partial t} = f\left(I_{(x,t)}, h_{(x,t)}\right) - \beta \cdot h(x,t) + D\frac{\partial^2 h(x,t)}{\partial x^2} \tag{1}$$

where $\beta$ and $D$ denote the degradation rate and diffusion constant of Hb, respectively, $h(x, t)$ denotes the concentration of Hb at time $t$ and the AP axis coordinate $x$, and $f(I,h)$ is the gene regulatory function (GRF) that determines the synthesis rate of Hb.

$I(x,t)$ represents the net regulation effect of the maternal factors, which are generally spatial-dependent and time-variant. The maternal factor Nos represses the translation of maternal *hb* (*Wang and Lehmann, 1991*), so we assume that Nos only affects the initial distribution of Hb, $h(x,0) = h_m \cdot S(k_x(x - x_0))$, where $S(\xi) = 1/(1 + \exp(\xi))$. Another maternal factor Bcd constantly activates the expression of Hb during early embryogenesis. Hence, we assume $I(x,t) = b(x,t)$, where $b(x,t)$ denotes the nuclear Bcd gradient. The nuclear Bcd gradient is dynamic and its amplitude decays after nc12 (*Little et al., 2011*). As an approximation, we assume

$b(x,t) = b_m \cdot e^{-x/\lambda} \cdot T(t; \omega_0, t_0)$, where $T(t; \omega_0, t_0) = \begin{cases} 1 & , t \leq t_0 \\ \exp(-\omega_0(t - t_0)) & , t_0 \end{cases}$, denoting that the Bcd profile decays with the linear decay rate $\omega_0$ in an isotropic manner after $t_0$ (time offset from the onset of nc14). Besides the activation from Bcd, Hb is also activated by itself (*Lopes et al., 2008*). Considering that the P2 enhancer of *hb* has at least six binding sites for Bcd (*Driever et al., 1989*) and three for Hb (*Treisman and Desplan, 1989*), and given the high binding cooperativity of Bcd or Hb, we assume that the GRF in Eq.1 takes the all-or-nothing strategy by following a coupled Hill function:

$$f(b,h) = \frac{\alpha_b \left(\frac{b}{b_0}\right)^{n_b} + \alpha_h \left(\frac{h}{h_0}\right)^{n_h}}{1 + \left(\frac{b}{b_0}\right)^{n_b} + \left(\frac{h}{h_0}\right)^{n_h}} \tag{2}$$

where $b_0$ and $h_0$ are the activation thresholds for Bcd and Hb, respectively, $n_b$ and $n_h$ are the Hill coefficients for Bcd and Hb, respectively, and $\alpha_b$ and $\alpha_h$ denote the scaling factor for the production of Hb from the activation of Bcd and the self-activation of Hb, respectively.

The experiment showed that *stau⁻* mutants are different from the WT in Bcd gradients and maternal Hb gradients: the amplitude of Bcd gradients of *stau⁻* mutants ($b_m^{stau-}$)) is ~35% of that of the WT ($b_m^{wt}$)); the length constant of Bcd gradients of *stau⁻* mutants ($\lambda^{stau-}$)) is ~17% larger than that of the WT ($\lambda^{wt}$)); and the initial distribution of Hb of *stau⁻* mutants is uniform across the embryo due to the lack of repression from the Nos gradient. Hence, we set all the parameters of *stau⁻* mutants to be the same as those for the WT except for $b_m^{stau-} = 0.35 \cdot b_m^{wt}$, $\lambda^{stau-} = 1.17 \cdot \lambda^{wt}$, and (c) $x_0^{stau-} \to +\infty$.

This model has a total 16 parameters, nine of them are fixed on the basis of the experimental values from references or our measurements, and the others need to be optimized from data fitting (*Figure 6—source data 1*). To obtain seven free parameters, we fit the simulated $x_{hb}$ as a function of $t$ to the experimental dynamics of $x_{hb}$ for both *stau⁻* mutants and the WT. The error function weighted during the early rising phase of the WT, the final position difference between *stau⁻* mutants and the WT, and the final changing rate of $x_{hb}$ helps to capture the key dynamic characteristics and to avoid potential over fittings.

Using optimized parameters, we simulated the dynamics of $x_{hb}$ for Bcd1.0 by setting $b_m^{1x} = 0.5 \cdot b_m^{wt}$. We also simulated $x_{hb}$ on the ventral side by setting $b_m^{ventral} = 0.62 \cdot b_m^{dorsal}$ and $\lambda^{ventral} = 1.1 \cdot \lambda^{dorsal}$ to obtain the best fitting to the experimental results, that is, 6% EL and 3% EL for *stau⁻* mutants and the WT, respectively (*Figure 6—figure supplement 1B*). To test the effect of the dynamics of Bcd gradients on the formation of the Hb boundary, we set $\omega_0 \to 0$ and ran the simulation using the same optimized parameters or ran the parameter optimization again to obtain the best fit.

## Acknowledgements

We thank John Reinitz for antibodies, Thomas Gregor and Yi Rao for fly lines, and Lu Wang for assistance with the *Drosophila* experiment. This project is supported by the National Natural Science Foundation of China 31670852 and the 100-talent plan of Peking University. The *Drosophila* lab used in this project is supported by Peking-Tsinghua Center for Life Sciences. The modeling optimization was performed on the High Performance Computing Platform of the Center for Life Sciences, Peking University.

## Additional information

### Funding

| Funder | Grant reference number | Author |
| --- | --- | --- |
| National Natural Science Foundation of China | The General Program 31670852 | Feng Liu |
| Peking University | 100-talent plan | Feng Liu |

The funders had no role in study design, data collection and interpretation, or the decision to submit the work for publication.

## Author contributions
Zhe Yang, Hongcun Zhu, Data curation, Investigation, Visualization, Methodology; Kakit Kong, Software, Formal analysis, Investigation, Visualization, Modeling; Xiaoxuan Wu, Jiayi Chen, Peiyao Li, Data curation, Investigation; Jialong Jiang, Jinchao Zhao, Software, Investigation, Modeling; Bofei Cui, Data curation, Software, Investigation, Visualization, Methodology; Feng Liu, Conceptualization, Resources, Data curation, Software, Formal analysis, Supervision, Funding acquisition, Validation, Investigation, Visualization, Methodology, Writing - original draft, Project administration, Writing - review and editing

## Author ORCIDs
Feng Liu (iD) https://orcid.org/0000-0001-9724-6127

## Decision letter and Author response
Decision letter https://doi.org/10.7554/eLife.54276.sa1
Author response https://doi.org/10.7554/eLife.54276.sa2

# Additional files

## Supplementary files
• Source code 1. 3D imaging analysis code.

• Transparent reporting form

## Data availability
A representative set of 3D imaging data reported in this paper has been deposited in the Dryad repository, http://dx.doi.org/ (doi. 10.5061/dryad.mcvdncjxw). All the other data generated or analysed during this study are included in the manuscript and supporting files.

The following dataset was generated:

| Author(s) | Year | Dataset title | Dataset URL | Database and Identifier |
|---|---|---|---|---|
| Liu F, Yang Z, Zhu H, Kong K, Wu X, Chen J, Li P, Jiang J, Zhao J, Cui B | 2020 | The dynamic transmission of positional information in stau-mutants during Drosophila embryogenesis | https://dx.doi.org/10.5061/dryad.mcvdncjxw | Dryad Digital Repository, 10.5061/dryad.mcvdncjxw |

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
