## [Decision Letter]

**Acceptance summary:**

The paper describes a new observation in early *Drosophila* development, namely that the expression boundaries of certain gap genes temporarily shift position in a mutant which lacks the important protein Staufen. From this the authors infer that initial stages of boundary positioning are consistent with a threshold-like model, but later stages exhibit self-organized patterning. This result opens up new questions about the role of Staufen, and we hope will trigger future work to unravel the precise mechanism of action of Staufen in *Drosophila* development.

**Decision letter after peer review:**

Thank you for sending your article entitled "The dynamic transmission of positional information in *stau*^–^ mutants during *Drosophila* embryogenesis" for peer review at *eLife*. Your article is being evaluated by two peer reviewers, and the evaluation is being overseen by a Reviewing Editor and Aleksandra Walczak as the Senior Editor.

As you can see from the reviews below, the reviewers found that your observations of the shift in gap gene expression boundaries in *stau*^–^ mutants are potentially interesting and could change our understanding of the role of Staufen. However, they both felt that much more analysis and additional experiments are required to properly understand the difference between the mutant and the wild-type.

Reviewer #1:

Early *Drosophila* development is remarkably robust. Gene expression boundaries are defined with single cell accuracy along the embryo. Understanding how such precision is achieved remains a major challenge. Staufen is a key gene in this process, with previous work highlighting that *stau*^–^ mutants have significant effects on Hunchback boundary precision.

In this manuscript, Yang et al. develop 3D quantitative approaches to probe boundary position in *stau^–^* mutants. They demonstrate that the gap gene expression boundaries are still precise, in the sense that at the same developmental time there is little variation between embryos. Yet, in *stau*^–^ mutants the boundaries of Hunchback and other gap genes shift significantly in time. Initial positions are quite distinct from wild-type embryos, but by gastrulation the boundary positioning is similar.

They use these results to directly test two competing hypotheses in embryonic patterning: threshold dependent positional information vs. self-organization. They draw the conclusion that initial stages of boundary positioning are consistent with a threshold-like model, but later stages display hallmarks of self-organized patterning.

The question tackled and the approaches used are both interesting, and certainly suitable for *eLife* in principle. However, I do have some concerns that I outline below.

1) One of my main concerns with the manuscript is the details of the projection. The authors present data that looks excellent, but only as a heat map. Details of the projection procedure are poor and not convincing. Why not use establish protocols that robustly segment a curved object, such as Heemskerk and Streichan, 2015?

2) Relatedly, in the Materials and methods the authors do a single 180º rotation of the embryo. Therefore, even using a light-sheet, I do not see how they image the lateral sides of the embryo well – they pick up the ventral and dorsal. There is significant decrease in image quality on a light-sheet with deeper penetration, particularly on a Zeiss Z1. The authors do not show clear images of the raw data. Related to point 1, I don't see how such smooth projections can be achieved with the microscopy protocol outlined in the Materials and methods. The authors need to (i) show raw data; (ii) "raw" data after image combining; (iii) clearer method of how the projected image is converted into the heat maps shown in the figures.

3) Again, related to the imaging, but light-sheet data is notoriously noisy. For example, there is striping and artefacts leave clear marks etc. Yet, the data looks very clean. If anything, this noise would increase the measured errors (so not really going against the authors’ central claims). But, more detail is needed in how the analyzed data was curated and imaging artefacts corrected for.

4) The description of the behavior of the Hunchback boundary in *stau* needs to be clearer. There are two competing issues. First, at a specific time point, the embryo-to-embryo variability is small and comparable to wildtype. Second, the boundary itself shifts in time in a reproducible manner during n.c. 14. At the moment, in both the Abstract and Introduction paragraph six, the attempt to combine both of these ideas in a single sentence is extremely confusing (reading the text, I didn't understand what was meant – only the figures clarified for me what the authors were trying to communicate). The authors need to find a wording that brings out the key ideas more clearly.

5) Given that the final Hb boundaries aren't that perturbed in the *stau* mutants, I'm confused why there are only four Eve stripes. Can the authors use their model to, at least, hypothesize why three Eve boundaries are lost in *stau* mutants?

6) I agree it's good to generate clean mutants. However, validation of the CRISPR mutant should be shown along with evidence that there are no off-target effects. It is not sufficient to say "data not shown" in the Materials and methods.

7) When measuring precision, a large n is needed to be confident of measured errors. However, some of the presented data represents only 5 embryos (I believe). A power-analysis should be performed to find the number of embryos required to make a statistically robust conclusion. I would be very surprised if 5 were sufficient. Also, in general, in every figure legend, the number of embryos used in each panel should be clearly stated.

8) In the normalization of the Hb, why not normalize to the posterior peak? This is independent of Bicoid and provides a less biased readout for normalization. In this case, how are the results altered?

9) All the data presented is assuming scaling of the system. Are there scaling defects in *stau* mutants? The authors should present, at least in a figure supplement, scaled and unscaled data so a comparison can be made.

Reviewer #2:

The paper has a single interesting observation but would need much more extensive analyses for publication in *eLife*.

The major new result is that the HB boundary in Staufen mutants shifts position to the posterior during cycle 14 (from 36 to 47% EL). Although wild type embryos also show a posterior shift, it is much smaller, from 45 to 48 EL. Since the boundary position in mutant and wildtype are ultimately very similar, the major defect in the mutants seems to be an HB expression that is initially positioned too far anterior but then corrects later during cellularization.

The result is interesting in that earlier studies had observed an increase variability in the position of the Hb boundary in Staufen mutant, but had attributed it to a role for wild type Staufen product in error correction and precision. Since those earlier studies defined Hb boundaries in collections of embryos that were not as carefully timed nor as carefully oriented as the present study, the new observations in suggests that the fundamental defect in Staufen mutants is not an increased error or variability, but an altered dynamic behavior, a behavior that when summed mimics increased error.

This re-definition of the Staufen phenotype is an important observation, and might provide an interesting entry point for an exciting analysis connecting Staufen function to the initiation of Hb expression. In my view, such further analyses are essential for publication in *eLife*. The authors need to explain how this new better-defined phenotype arises and how it relates to the wildtype role of Staufen in patterning.

Although the remaining portion of the Results section presents data about other features of the phenotype (e. g., the reproducibility and slope of the Bcd gradient), in most of this data Staufen and wildtype behave similarly and thus don't provide insights into what is fundamentally different in the mutants. The Discussion presents a model that reproduces some features of the Staufen phenotype, but no new data is presented specifically in support of that model.

Regardless of whether and where the experiments are ultimately published, two is issues need to be clarified:

1) In their characterization of Staufen mutants, the authors describe a second feature of the HB boundary not mentioned in in the earlier studies, namely in Staufen mutants its position on the ventral side is approximately 6% farther anterior on the ventral side at all stages than it is on the dorsal side (30 to 37 at early stages and 41 to 47 at later stage). There does not seem to be a dynamic component in this DV difference, the boundary shifts posteriorly by 11% but maintains the same DV slant. It is presumably an inadvertent omission, but the authors do not present data on whether a comparable slant is observed in wild type. If it is a Staufen specific feature, then the DV difference is a new and potentially informative phenotype that needs to be incorporated into models for Staufen function.

2) The authors go to considerable lengths to generate a new version of Staufen HL54 for these studies using CRISPR-Cas9, although this is only mentioned in the Discussion. This is the allelic variant used in all the earlier studies and it is interesting that the new and old versions of the allele behave similarly, although the authors’ new data suggests an alternate interpretation for the measured variability. In fly base, the HL54 allele is described as "hypomorphic" with partial loss of function. Is it possible that the earlier describe "variability" and the abnormal early expression described in the current manuscript is an allelic specific feature of the partial loss of function? The authors need to show whether true loss of function alleles show the same phenotype. Such alleles exist (e. g., *stau^D3^*) and have actually been used more frequently in analyses of Staufen's role in various developmental processes.

[Editors' note: further revisions were suggested prior to acceptance, as described below.]

Thank you for resubmitting your work entitled "The dynamic transmission of positional information in *stau*^–^ mutants during *Drosophila* embryogenesis" for further consideration by *eLife*. Your revised article has been evaluated by Aleksandra Walczak (Senior Editor) and a Reviewing Editor.

The manuscript has been improved but there are some remaining issues that need to be addressed before acceptance, as outlined below:

Please address the major issue raised by reviewer 1. From reviewer 2's comments, please address the issue raised about parameter values and distributions by clearly mentioning which parameter values (or ranges) are drawn from experimental data, with references, and which are unknown. It seems the most significant aspect of the new description of the Staufen phenotype is the self-correcting shift. The fact that the explanation for this phenomenon provided does not apply to, say Hb, suggests that Staufen is somehow different, or that things don't work the way the authors' model suggests. This needs to be made very explicit in the text.

Reviewer #1:

Overall, the paper has been substantially improved. The experiments are presented in a more rigorous manner, with care taken to explain the uncertainties. The discussion of the *stau* mutants is also stronger.

The main argument regarding the effect of *stau* on Hb precision is very interesting and of potential broad impact. Though the current paper does not conclusively demonstrate the specific pathway of action, I feel that the evidence presented, combined with the model, is sufficient for publishing. Hopefully, the authors and others will build on this work to really dissect the mode of action of Stau in the early embryo.

I have one major issue. The authors claim that the increased Bicoid decay length is "reasonable because the *bcd* mRNA is more extensively distributed in the embryos of *stau*^–^". However, in the SDD model of Bicoid gradient formation, changing the source distribution would not alter the actual decay length. The experimentally measured decay length is an amalgamation of the diffusion and degradation terms (which define the "real" decay length) and the distributed source of *bcd* mRNA. Therefore, the current wording is not precise. Further, it is straightforward to test the authors hypothesis; take the SDD model for different source terms and see how changing the source distribution alters the "measured" decay length.

Reviewer #2:

The revised manuscript addresses and clarifies many of the quantitative issues raised by the reviewers, but still fails to address what I believe to be the fundamental challenge. In the original manuscript, the authors provided a new description of the Staufen phenotype, arguing against a previously suggested role governing the variability of Hunchback expression. Instead, mutant embryos show an initial precise anteriorly shifted HB expression pattern that then self-corrects to a more normal expression pattern during cycle 14.

I believe that for publication in *eLife*, the authors need use this new phenotype to better understand the role of Staufen. Staufen's well described and previously published role in Bcd RNA localization provides a simple explanation of the initial anterior position of the boundary. The challenge is to explain its subsequent posterior shift and the other new features of the phenotype described by the authors. Here the manuscript fails. Thus while a better description of the Staufen phenotype is a useful contribution to the field, the manuscript does not meet the high standards expected for *eLife* papers.

There is very little new data or experiments in the revised manuscript. The authors generate a second mutant allele that mimics a previously describe null mutation in Staufen (stau^D3^) and use it to confirm that the new phenotypes are not allele specific variants. The remaining revisions supply more detailed or rigorous analyses of the data, and use mathematical models to test for internal consistency. The models requires knowledge of distributions and parameters that are not always available, and even when they provide outcomes consistent with the observations (e. g., the DV slant of the boundary), they don't really link those effects mechanistically to the known molecular function of Staufen.

A central question remains – why the expression boundary of HB in Staufen mutants shifts to an almost normal position by late cycle 14, when similar corrective shifts is not observed following other genetic manipulations of Bcd that alter the HB boundary. The authors attribute the shift in Staufen to the self-activation of Hb and the cross-regulation from the other gap genes but do not explain those processes do not occur in the other cases. Staufen differs from the other Bicoid manipulations in that it also affects the distribution of Nanos and thus maternal Hb. How this additional role would affect the shifting process is not clear, especially given that in classic papers, the osk/nanos pathway is not thought to affect HB boundary position very much at all (see for example the original analyses of Driever and Nüsslein-Volhard). The authors' suggestion might point to the existence of previously undetected interactions between the two systems, but in the absence of new data and more detailed measurements, the suggestion itself does not merit publication in *eLife*.

Overall the manuscript provides an interesting entry point into a fascinating problem, but needs more serious development and data for it to have the impact it deserves.

---

## [Author Response]

Reviewer #1:[…]1) One of my main concerns with the manuscript is the details of the projection. The authors present data that looks excellent, but only as a heat map. Details of the projection procedure are poor and not convincing. Why not use establish protocols that robustly segment a curved object, such as Heemskerk and Streichan, 2015?2) Relatedly, in the Materials and methods the authors do a single 180º rotation of the embryo. Therefore, even using a light-sheet, I do not see how they image the lateral sides of the embryo well – they pick up the ventral and dorsal. There is significant decrease in image quality on a light-sheet with deeper penetration, particularly on a Zeiss Z1. The authors do not show clear images of the raw data. Related to point 1, I don't see how such smooth projections can be achieved with the microscopy protocol outlined in the Materials and methods. The authors need to (i) show raw data; (ii) "raw" data after image combining; (iii) clearer method of how the projected image is converted into the heat maps shown in the figures.3) Again, related to the imaging, but light-sheet data is notoriously noisy. For example, there is striping and artefacts leave clear marks etc. Yet, the data looks very clean. If anything, this noise would increase the measured errors (so not really going against the authors central claims). But, more detail is needed in how the analyzed data was curated and imaging artefacts corrected for.

Thank you very much for your great suggestions. We describe our method of 3D reconstruction and projection in more detail in the Materials and methods and will deposit the image analysis code online upon publication. We compare our projection method with one of the well-established protocols by Berkeley *Drosophila* Transcription Network Project (Fowlkers, et al., 2008) and confirm that the final heat maps from the reconstructed 3D embryos are the same. We also test the Heemskerk and Streichan’s method with our data and find that their heat map can show higher resolution along y-axis than ours, which is limited by the angle step of the projection. But for the task of extracting the expression profile, the two methods are still comparable with each other (Figure 1—figure supplement 1G-I).

We add the typical raw images taken with the light-sheet microscope, after imaging combination, and after projection at different angles in the supplementary figures. We plan to deposit a set of raw data online upon publication. Based on the raw images, the fluorescence intensity is decaying as the penetration depth increases. But the fixed embryo shrinks and its imaging depth is less than 160 μm. Hence the intensity decrease is not very severe. Moreover, we rotate the embryos by 180^o^ and take the image stacks from both directions, which complement the intensity for each other. We then fuse the two stacks after image registration and confirm that the intensity is much more uniform along with the penetration depth (Figure 1—figure supplement 1F). So the image quality of the projected plane is comparable along the AP axis from different embryo orientations including dorsal, ventral and lateral view.

Although it is well known that for the large, semi-opaque biological samples contain strong scattering or absorbing features, stripping-like artifacts may be observed, we have not observed severe distortion caused by this problem. We find that the key to improve image quality includes: (i) Light sheet illumination setting: chose the “Dual Side when Experiment” and the “Online Dual Side Fusion”, then adjusted the left and right beam path to get the optimal images; (ii) Pivot scan setting: the “Pivot scan” was activated to reduce the shadows which might otherwise be cast by optically dense structures within the embryos.

To clarify these issues, we revised the “3D imaging and analysis” section in the Materials and methods, updated Figure 1—figure supplement 1, and added several sentences in:

“However, if strong scattering or absorbing objects exist in samples, one needs to take caution to alleviate or properly correct the potential striper shadow artifacts (Mayer et al., 2018). The conventional 2D expression profiles can be extracted with 3D imaging analysis tools developed by us and the others (Fowlkes et al., 2008; Heemskerk and Streichan, 2015).”

4) The description of the behavior of the Hunchback boundary in stau needs to be clearer. There are two competing issues. First, at a specific time point, the embryo-to-embryo variability is small and comparable to wildtype. Second, the boundary itself shifts in time in a reproducible manner during n.c. 14. At the moment, in both the Abstract and Introduction paragraph six, the attempt to combine both of these ideas in a single sentence is extremely confusing (reading the text, I didn't understand what was meant – only the figures clarified for me what the authors were trying to communicate). The authors need to find a wording that brings out the key ideas more clearly.

We changed the description in the Abstract to be:

“With improved control of measurement errors, we show *x_Hb_*of *stau^–^*mutants reproducibly shifts posteriorly by 10% of the embryo length (EL) to the wild type (WT) position in the nuclear cycle (nc) 14, and its variability at short time windows is comparable as that of the WT.”

Furthermore, we revised the Results as follows:

“Although *x_Hb_*dynamically shifts posteriorly by more than 10% EL in *stau^–^*mutants, this shift is reproducible from embryo to embryo. Within a short time window, the variability of *x_Hb_*in *stau^–^*mutants is comparable with that of the WT (Figure 1E). Furthermore, even for all the data in the whole nc14, after de-trending the dynamic shift, the standard deviations of *x_Hb_*for *stau^–^*mutants is 1.67 ± 0.16% EL (errors are estimated with bootstrap), similar to 1.45 ± 0.14% EL for the WT. Moreover, the variance difference between the two fly lines is not statistically significant (*p* = 0.31 in a two-sample *F*-test for equal variances) (Figure 1F-G).”

5) Given that the final Hb boundaries aren't that perturbed in the stau mutants, I'm confused why there are only four Eve stripes. Can the authors use their model to, at least, hypothesize why three Eve boundaries are lost in stau mutants?

The loss of three Eve peaks is a known phenotype of *stau^–^*mutants, just like *nos^–^*mutants (Lehmann and Nusslein-Volhard, 1991). This is reasonable, as the Nos gradient is also abolished in both mutants. A recent paper from Thomas Gregor lab proposed a framework to decode the pair-rule gene profiles from four gap gene profiles and give an excellent prediction on the Eve profiles of *nos^–^*mutants (Petkova, et al., 2019). Hence our manuscript only focuses on modelling the origin of the large shift of the Hb boundary. In the future work, we plan to investigate how the large dynamic shift affects the expression of the other gap genes and downstream pair rule genes with modeling.

6) I agree it's good to generate clean mutants. However, validation of the CRISPR mutant should be shown along with evidence that there are no off-target effects. It is not sufficient to say "data not shown" in the Materials and methods.

We showed the images of the cuticle samples of the CRISPR mutant as Figure 4—figure supplement 3. And we confirm that they are very similar to the published *stau^–^*mutants.

7) When measuring precision, a large n is needed to be confident of measured errors. However, some of the presented data represents only 5 embryos (I believe). A power-analysis should be performed to find the number of embryos required to make a statistically robust conclusion. I would be very surprised if 5 were sufficient. Also, in general, in every figure legend, the number of embryos used in each panel should be clearly stated.

We agree with you that the sample size is indeed an important consideration. The sample size choices for our study were informed by the following: (i) previous gap gene precision measurements, (ii) cost and time availability of sample acquisition, and (iii) power calculations. First, in previous measurements on gap gene boundary/peak precision in fly embryos in nc14, the total sample size is usually less than 100 and the sample size of each bin in short time windows is less than 10 (Thomas, et al., Cell, 2007; Heng, et al., Development Cell, 2008; Aitana, et al., Molecular Systems Biology, 2010; Julien, et al., Molecular Systems Biology, 2013; Feng, et al., PNAS, 2013; Honggang, et al., Nature Communications, 2015). All of these measurements used the conventional method with difficulty in controlling the embryo orientations. Second, the new method with 3D imaging could reduce the measurement error related to embryo orientations, but even with the fast imaging light sheet microscope, it takes much longer time per embryo. So it is time-consuming and costly to obtain a large sample size. Third, our choice of sample sizes was also guided by the power analysis based on previous studies. For the WT, the variability of *x_hb_*is approximately 1 ± 0.5% EL (the error represents the standard deviation from boots-trapping, Julien, et al., Molecular Systems Biology, 2013). As for *stau^–^*mutants, the variability of *x_hb_*is 6% EL (Houchmandzadeh, et al., 2002) or 2.78% EL (Heng, et al., Development Cell, 2008), hence we estimate its variability as 4 ± 2% EL. Specificity, we evaluate the power of detecting the difference of the variability of *x_hb_*between the WT and *stau^–^*mutants. Our computation, under alpha level. 05 and power (1-β) = . 8, indicated a sample size of 5 embryos is adequate.

In the original manuscript, we measured total 40 *stau^–^*mutant embryos and 34 WT embryos. These samples were divided into 6 bins and the sample size in each bin is as least 5. Moreover, to pool all the measured data without binning, we apply the de-trend analysis to remove the effect of the average dynamic shift on the variability of *x_hb_*. We find that the standard deviations of the residual from a smooth spline fit is 1.85% EL (*n*=40) and 1.41% EL (*n*=34) for *stau^–^*mutants and the WT, respectively. Two-sample *F*test for equal variances cannot be rejected (*p* = 0.1). This is consistent with our binning results shown in Figure 1E.

Nevertheless, in order to improve the temporal resolution in precision measurements, we further increase the sample size of WT to *n*=47, and *stau^–^*mutants to *n*=69. We update all the related figures and corresponding statistics. We add the de-trend analysis result as Figure 1F-G, and mark the sample number in each figure caption in the revision.

8) In the normalization of the Hb, why not normalize to the posterior peak? This is independent of Bicoid and provides a less biased readout for normalization. In this case, how are the results altered?

Although the posterior peak is independent of Bicoid, it only shows up late in nc14. Hence it is very noisy to use it as the normalization reference in early nc14. Nevertheless, for the Hb profiles in late nc14, we compare this new normalization method with the conventional one, i.e., normalize the Hb profile to the maximum intensity of the whole profile. And we confirm that the difference of *x_hb_*is within the measurement errors.

**Author response image 1. respfig1:** Comparison between conventional normalized method and the normalization using the posterior peak as the reference. (**A–B**) Normalized profiles of Hb intensity of *stau*^–^ mutants in one developmental time window (45-50 min into nc14, sample number: n = 5) using the conventional method (**A**) and posterior peak normalization method (**B**). Different colors represent different samples. (**C**) The average profile of Hb intensity with the conventional method (A, blue) and posterior-peak-normalization method (B, red). The boundary positions are 47.4% EL (blue) and 47.8% EL (red), respectively.

9) All the data presented is assuming scaling of the system. Are there scaling defects in stau mutants? The authors should present, at least in a figure supplement, scaled and unscaled data so a comparison can be made.

Thank you for your insightful suggestion. The Hb boundaries of *stau^–^*mutants are still scaling with the embryo length. And both the slope and *R2* in the fitting of *stau^–^*mutants are very similar to those of the WT. Since *x_hb_*dynamically shifts in a great range for *stau^–^*mutants, this scaling effect of *stau^–^*mutants is significant only if we normalize the absolute Hb boundary position of each embryo by the average position in the corresponding developmental time. To clarify this issue, we added Figure 1H-I and a paragraph in the text:

“The variability of *x_Hb_*also depends on whether it is scaling with the embryo length (Houchmandzadeh et al., 2002; He et al., 2008). […] Without this normalization, the scaling of the absolute Hb boundary position of *stau^–^*mutants (*R2* = 0.26) is inferior compared with the WT (*R2* = 0.71), consistent with previous results (He et al., 2008).”

Reviewer #2:The paper has a single interesting observation but would need much more extensive analyses for publication in eLife.The major new result is that the Hb boundary in Staufen mutants shifts position to the posterior during cycle 14 (from 36 to 47% EL). Although wild type embryos also show a posterior shift, it is much smaller, from 45 to 48 EL. Since the boundary position in mutant and wildtype are ultimately very similar, the major defect in the mutants seems to be an Hb expression that is initially positioned too far anterior but then corrects later during cellularization.The result is interesting in that earlier studies had observed an increase variability in the position of the Hb boundary in Staufen mutant, but had attributed it to a role for wild type Staufen product in error correction and precision. Since those earlier studies defined Hb boundaries in collections of embryos that were not as carefully timed nor as carefully oriented as the present study, the new observations in suggests that the fundamental defect in Staufen mutants is not an increased error or variability, but an altered dynamic behavior, a behavior that when summed mimics increased error.This re-definition of the Staufen phenotype is an important observation, and might provide an interesting entry point for an exciting analysis connecting Staufen function to the initiation of Hb expression. In my view, such further analyses are essential for publication in eLife. The authors need to explain how this new better-defined phenotype arises and how it relates to the wildtype role of Staufen in patterning.Although the remaining portion of the Results section presents data about other features of the phenotype (e. g., the reproducibility and slope of the Bcd gradient), in most of this data Staufen and wildtype behave similarly and thus don't provide insights into what is fundamentally different in the mutants. The Discussion presents a model that reproduces some features of the Staufen phenotype, but no new data is presented specifically in support of that model.

We agree with you that it is very important to connect the re-defined *stau^–^*phenotype with the *stau* function in the initiation of Hb expression. Based on our experimental results, the activation of Hb at an early developmental time is consistent with the conventional threshold dependent positional information model. Without Stau, the localization of the *bcd* mRNA at the anterior pole cannot be sustained, hence the amplitude of the Bcd gradient decreases by 65% and the length constant increases by 17% in *stau^–^*mutants. As a result, the initial Hb position shifts by ~10% EL from the WT position. But as embryo development progresses, the self-activation of Hb and the cross-regulation from the other gap genes shift the Hb boundary position towards the WT position. To clarify this issue, we extend the model to explain the origin of the large shift of the Hb boundary in *stau^–^*mutants, e.g., considering the dynamics of the Bcd gradient, and rewrote the Discussion part on the modeling to illustrate more clearly the connection between the Stau function and the regulation of Hb expression:

“To test this idea, we constructed a mathematical model to calculate the dynamic shift in *x_Hb_*in both *stau^–^*mutants and the WT. […] This model fits well with the measured dynamic shift of *x_Hb_*in both *stau^–^*mutants and the WT (Figure 6D), indicating that the synergy effect resulting from the altered Bcd gradients and maternal Hb in *stau^–^*mutants can account for the much larger shift of *x_Hb_*compared with the WT.”

Regardless of whether and where the experiments are ultimately published, two is issues needed to be clarified1) In their characterization of Staufen mutants, the authors describe a second feature of the Hb boundary not mentioned in in the earlier studies, namely in Staufen mutants its position on the ventral side is approximately 6% farther anterior on the ventral side at all stages than it is on the dorsal side (30 to 37 at early stages and 41 to 47 at later stage). There does not seem to be a dynamic component in this DV difference, the boundary shifts posteriorly by 11% but maintains the same DV slant. It is presumably an inadvertent omission, but the authors do not present data on whether a comparable slant is observed in wild type. If it is a Staufen specific feature, then the DV difference is a new and potentially informative phenotype that needs to be incorporated into models for Staufen function.

Thank you very much for your insightful comments. This is indeed a very interesting and important phenotype of *stau^–^*mutants. In the WT, our data shows that the difference of *x_hb_*in the dorsal side and ventral side remains approximately 3% EL, smaller than that of *stau^–^*mutants. We added a sentence in the text:

“Moreover, *x_Hb_*differs significantly in different orientations, e.g., the ventral boundary moves from 32.7 ± 2.4% EL by 10.9% EL to 43.6 ± 1.2% EL, shifting anteriorly by approximately 6% EL compared with the dorsal boundary (Figure 1D and Figure 1—figure supplement 4A, D, and F). In contrast, the difference of *x_Hb_*between the dorsal and ventral sides in the WT is only 3% EL (Figure 1D).”

To find the origin of this DV difference on *x_hb_*, we analyze the difference of the Bcd gradient on the dorsal side and the ventral side. Based on our live imaging experiment, the ventral Bcd gradient shows decreased amplitude by 24% but increased length constant by 10% compared with the dorsal Bcd gradient. Moreover, if we apply this difference, our updated model can replicate the DV difference of *x_hb_*in both *stau^–^*mutants and the WT when the amplitude adjustment is changed to 38%. We speculate that this discrepancy might result from the DV difference in the maturation correction curve of Bcd-GFP. To clarify this issue, we added Figure 6—figure supplement 1B-C and the following sentences:

“Moreover, this model predicts that the initial position of *x_Hb_*varies as the Bcd gradient is tuned. Our measurement shows that Bcd gradients on the ventral side decrease in the amplitude and increase in the length constant compared with those on the dorsal side (Figure 6—figure supplement 1B), consistent with previous results (Gregor et al., 2007a). This difference could attribute to the observed anterior shift of *x_Hb_*on the ventral side, and the shift amount is approximately 6% EL for *stau^–^*mutants but 3% EL for the WT (Figure 6—figure supplement 1C).”

2) The authors go to considerable generate a new version of Staufen HL54 for these studies using CRISPR-Cas9, although this is only mentioned in the Discussion. This is the allelic variant used in all the earlier studies and It is interesting that the new and old versions of the allele behave similarly, although the authors new data suggests an alternate interpretation for the measured variability. In fly base, the HL54 allele is described as "hypomorphic" with partial loss of function. Is it possible that the earlier describe "variability" and the abnormal early expression described in the current manuscript is an allelic specific feature of the partial loss of function? The authors need to show whether true loss of function alleles show the same phenotype. Such alleles exist (e.g., stau^D3^) and have actually been used more frequently in analyses of Staufen's role in various developmental processes.

Thank you very much for your insightful comments. We prepared the fly line *stau^D3^*with CRISPR technique, confirmed that it shows the correct phenotype in the cuticle patterns (Figure 2). We measured the Hb profiles and confirmed that *stau^D3^*also shows a large posterior shift like *stau^HL54^*. We added these results in Figure 1—figure supplement 5, and added some comments in a paragraph in the text:

“Besides *stau^HL54^,* another *stau^–^*mutant, *stau^D3^*, also shows a large shift of *x_Hb_*: 42.8 ± 1.5% to 47.9 ± 1.8% from 12.5 min to 52.5 min into nc14 on the dorsal side (Figure 1—figure supplement 5). It is well known that *stau^D3^*is a strong allele with fully penetrant abdominal segmentation phenotype (Lehmann and Nüsslein-Volhard, 1991), hence the observed large shift of *x_Hb_*in *stau^–^*mutants is not an allelic specific feature for *stau^HL54^,* which could be hypomorphic, i.e., partial function loss.”

[Editors' note: further revisions were suggested prior to acceptance, as described below.]

Reviewer #1:Overall, the paper has been substantially improved. The experiments are presented in a more rigorous manner, with care taken to explain the uncertainties. The discussion of the stau mutants is also stronger.The main argument regarding the effect of stau on Hb precision is very interesting and of potential broad impact. Though the current paper does not conclusively demonstrate the specific pathway of action, I feel that the evidence presented, combined with the model, is sufficient for publishing. Hopefully, the authors and others will build on this work to really dissect the mode of action of Stau in the early embryo.I have one major issue. The authors claim that the increased Bicoid decay length is "reasonable because the bcd mRNA is more extensively distributed in the embryos of stau^–^". However, in the SDD model of Bicoid gradient formation, changing the source distribution would not alter the actual decay length. The experimentally measured decay length is an amalgamation of the diffusion and degradation terms (which define the "real" decay length) and the distributed source of bcd mRNA. Therefore, the current wording is not precise. Further, it is straightforward to test the authors hypothesis; take the SDD model for different source terms and see how changing the source distribution alters the "measured" decay length.

Thank you very much for your great suggestion. We tested the SDD model for broad source terms and confirmed that the length constant of *stau*^–^ mutant increases compared with the WT. We added two new figures as Figure 3—figure supplement 1A-B and revised the related sentences as the following

“This apparent increase of the measured length constant could result from more extensively distributed *bcd* mRNA in the embryos of *stau^–^*mutants (Ferrandon et al., 1994; Petkova et al., 2014), consistent with the simulation based on the synthesis-diffusion-degradation (SDD) model (Grimm et al., 2010) (Figure 3—figure supplement 1 A-B).”

Reviewer #2:The revised manuscript addresses and clarifies many of the quantitative issues raised by the reviewers, but still fails to address what I believe to be the fundamental challenge. In the original manuscript, the authors provided a new description of the Staufen phenotype, arguing against a previously suggested role governing the variability of Hunchback expression. Instead, mutant embryos show an initial precise anteriorly shifted HB expression pattern that then self-corrects to a more normal expression pattern during cycle 14.I believe that for publication in eLife, the authors need use this new phenotype to better understand the role of Staufen. Staufen's well described and previously published role in Bcd RNA localization provides a simple explanation of the initial anterior position of the boundary. The challenge is to explain its subsequent posterior shift and the other new features of the phenotype described by the authors. Here the manuscript fails. Thus while a better description of the Staufen phenotype is a useful contribution to the field, the manuscript does not meet the high standards expected for eLife papers.There is very little new data or experiments in the revised manuscript. The authors generate a second mutant allele that mimics a previously describe null mutation in Staufen (stau^D3^) and use it to confirm that the new phenotypes are not allele specific variants. The remaining revisions supply more detailed or rigorous analyses of the data, and use mathematical models to test for internal consistency. The models requires knowledge of distributions and parameters that are not always available, and even when they provide outcomes consistent with the observations (e. g., the DV slant of the boundary), they don't really link those effects mechanistically to the known molecular function of Staufen.A central question remains – why the expression boundary of HB in Staufen mutants shifts to an almost normal position by late cycle 14, when similar corrective shifts is not observed following other genetic manipulations of Bcd that alter the HB boundary. The authors attribute the shift in Staufen to the self-activation of Hb and the cross-regulation from the other gap genes but do not explain those processes do not occur in the other cases. Staufen differs from the other Bicoid manipulations in that it also affects the distribution of Nanos and thus maternal Hb. How this additional role would affect the shifting process is not clear, especially given that in classic papers, the osk/nanos pathway is not thought to affect HB boundary position very much at all (see for example the original analyses of Driever and Nüsslein-Volhard). The authors' suggestion might point to the existence of previously undetected interactions between the two systems, but in the absence of new data and more detailed measurements, the suggestion itself does not merit publication in eLife.Overall the manuscript provides an interesting entry point into a fascinating problem, but needs more serious development and data for it to have the impact it deserves.

We agree with you that the central question here is the link between the new phenotype of “self-correction” of the Hb boundary in *stau^–^*mutants and the function of Stau. And *stau^–^*mutants are special as both the Bcd gradient and the Nos gradient are distorted. Our current model actually shows that the synergy of the distorted Bcd and Nos gradient is sufficient to account for the large shift of the Hb boundary in *stau^–^*mutants if we assume self-activated Hb is activated by Bcd according to the threshold dependent model at early nc14, without introducing any unknown interactions. And our model shows that the depletion of the Nos gradient by the *stau^–^*mutant is necessary for the observed larger shift of *x_Hb_*in *stau^–^*mutants. As it predicts a very small shift in Bcd1.0 when only the Bcd amplitude is decreased. It also predicts that the shift dynamics is similar to Bcd1.0 if the Nos gradient is rescued in *stau^–^*mutants.

But since the current model only considers the activation of the Bcd and self-activation of Hb, it could not explain the total shift in Bcd1.0, and the small shift of *nos^–^*mutants as shown in the classic papers mentioned above. We speculate the different phenotype of *nos^–^*mutants compared with *stau^–^*mutants could be attributed to the cross-regulation of the other gap genes acted strongly in the posterior half (likely the repression from Kr), or some unknown repressors (Hongtao Chen, et al., Cell, 2012). Notably, the initial position of the Hb boundary in *nos^–^*mutants is much closer to the WT position than the *stau^–^*mutant. It is also possible that previously undetected interactions between the Bcd and Nos systems could attribute to the large shift. To clarify the complete picture, we could establish a comprehensive model incorporating both the maternal factors and the gap genes to dissect the sophisticated role of Stau in regulating Hb patterning. This is, however, a very challenging task, probably beyond the scope of the current manuscript. Based on our test, nearly all the published gap gene network models (Jaeger, et al., 2004; Verd, et al., 2017; Dmitri Papatsenko and Michael Levin, PLoS ONE; Jinxiang Shen, et al., bioRxiv, 2018) fail to replicate the “self-correction” phenotype discovered here.

To clarify this issue, we added simulation results of the Nos rescued *stau*^–^ mutant in Figure 6D, and revised the related paragraph as:

“The depletion of the Nos gradient by the *stau^–^*mutant is necessary for the observed larger shift of *x_Hb_*in *stau^–^*mutants. […] The shift amount of of *x_Hb_*in Bcd1.0 is only 1% EL, smaller than the experimental value of 4% EL (Figure 6D).”

And added a new paragraph discussing the limits of the current model:

“Hence, the current model suggests that the ultra-large shift of *x_Hb_*in *stau^–^*mutants can be attributed to the distorted Bcd and Nos gradient due to the loss of the function of Stau, if self-activated Hb is activated by Bcd according to the threshold dependent model at early nc14. However, this simplified model without cross-regulation of the gap genes fails to predict the small shift of *x_Hb_*in *nos^–^*mutants (Houchmandzadeh et al., 2002; Petkova et al., 2019). Thus a comprehensive model incorporating both the maternal factors and the gap genes is still needed to dissect the sophisticated role of Stau in regulating Hb patterning.”